# SK-VQA: Synthetic Knowledge Generation at Scale for Training Context-Augmented Multimodal LLMs

Xin Su [* 1]   Man Luo [* 1]   Kris W Pan [2]   Tien Pei Chou [2]   Vasudev Lal [1]   Phillip Howard [3]

## Abstract

Multimodal retrieval augmented generation (RAG) plays a crucial role in domains such as knowledge-based visual question answering (KB-VQA), where external knowledge is needed to answer a question. However, existing multimodal LLMs (MLLMs) are not designed for context-augmented generation, limiting their effectiveness in such tasks. While synthetic data generation has recently gained attention for training MLLMs, its application for context-augmented generation remains underexplored. To address this gap, we introduce SK-VQA, a large-scale synthetic multimodal dataset containing over 2 million visual question-answer pairs, each associated with context documents containing information necessary to determine the final answer. Compared to previous datasets, SK-VQA contains 11× more unique questions, exhibits greater domain diversity, and covers a broader spectrum of image sources. Through human evaluations, we confirm the high quality of the generated question-answer pairs and their contextual relevance. Extensive experiments show that SK-VQA serves both as a challenging KB-VQA benchmark and as an effective training resource for adapting MLLMs to context-augmented generation. Our results further indicate that models trained on SK-VQA demonstrate enhanced generalization in both context-aware VQA and multimodal RAG settings. SK-VQA is publicly available via Hugging Face Hub.

## 1. Introduction

Recent advances in Multimodal LLMs (MLLMs) have extended the impressive capabilities of LLMs to the vision

---
[*]Equal contribution   [1]Intel Labs   [2]Amazon   [3]Thoughtworks. Correspondence to: Xin Su <xin.su@intel.com>, Man Luo <man.luo@intel.com>.

*Proceedings of the 42nd International Conference on Machine Learning*, Vancouver, Canada. PMLR 267, 2025. Copyright 2025 by the author(s).

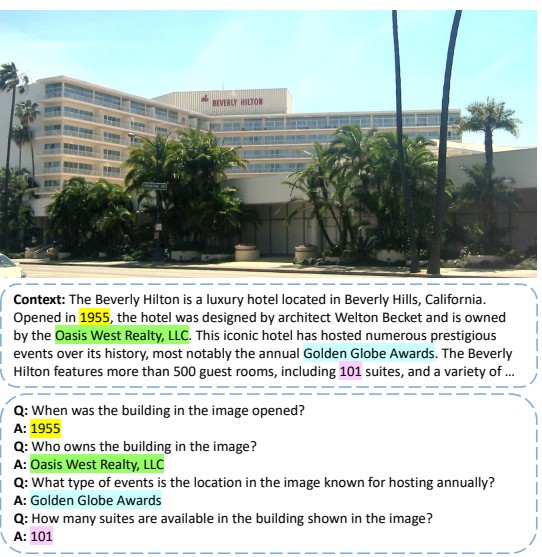

*Figure 1.* Example from SK-VQA. Each image is paired with a synthetically-generated context document & QA pairs.

domain, enabling advanced reasoning and chat capabilities over multimodal input queries consisting of both text and images (Achiam et al., 2023; Liu et al., 2024b). While MLLMs have demonstrated promising results, they suffer from the same hallucination and reliability issues as LLMs (Li et al., 2023; Zhou et al., 2023; Liu et al., 2024c). This motivates the need to incorporate MLLMs into retrieval-augmented generation (RAG) systems, where retrieved documents can ground answer generation in factually-correct information via context augmentation (Lewis et al., 2020; Ram et al., 2023). However, context augmentation for MLLMs presents unique challenges. Generated answers must be conditioned on both multimodal input queries as well as retrieved contexts which potentially span multiple modalities. Existing MLLMs have not been trained with context-augmented generation, which makes them incompatible for use in a RAG system. Adapting MLLMs for use in a RAG system requires extensive datasets which can support the training of models with multimodal queries and relevant context documents. Unfortunately, naturally-occurring data of this kind is relatively scarce; unlike other common types of internet data (e.g., text, image-text pairs), input queries consisting of both

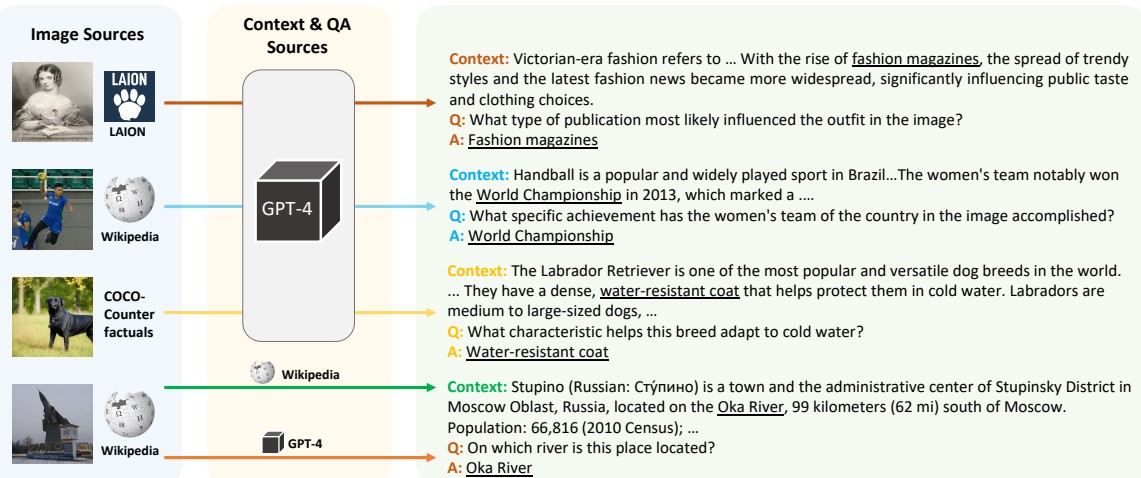

*Figure 2.* Examples of QA pairs in our dataset created for images and contexts collected from different sources. Unlike existing resources which are limited to images that can be linked to Wikipedia passages, our dataset can contain images from arbitrary sources by leveraging synthetically-generated context documents in addition to context documents sourced from Wikipedia.

images and text with associated context documents are not readily available at the scale needed for training MLLMs. This motivates us to develop a synthetic generation pipeline to create a large-scale, diverse dataset that facilitates the advancement of context-augmented MLLMs.

Synthetically generated data has recently grown in popularity as a solution for cases where sources of naturally-occurring data are scarce or have been exhausted in existing training datasets. In the context of training MLLMs, synthetic data has played an important role in the visual instruction tuning required to develop such models (Liu et al., 2024b). A limited number of knowledge-based visual question answering (KB-VQA) datasets suitable for training MLLMs in a context-augmented setting have been constructed by synthetically producing question-answer (QA) pairs for real images and related text documents (Chen et al., 2023; Lerner et al., 2022; Mensink et al., 2023). However, these existing resources are mostly limited to images which can be linked to context documents sourced from Wikipedia, focus on entity-specific knowledge, and lack question diversity due to their reliance on templates for constructing QA pairs. As we will demonstrate in the experiments (§5), training a model on such datasets leads to poor generalization.

To address these deficiencies, we construct SK-VQA: the largest KB-VQA dataset to-date, containing over 2 million QA pairs associated with synthetic context knowledge and images sourced from LAION (Schuhmann et al., 2021), WIT (Wikipedia images) (Srinivasan et al., 2021), and the synthetic COCO-Counterfactuals dataset (Le et al., 2024). Rather than relying on template-based methods to construct QA pairs for real data, we construct SK-VQA using a fully automated synthetic multimodal data generation approach which utilizes a strong foundation model (GPT-4) to produce relevant context documents and multiple QA pairs for a given image (Figure 1). This enables the acquisition of data which spans diverse sources of images (Figure 2), even allowing for the generation of fully-synthetic data which includes synthetic images, contexts, and QA pairs.

We show that SK-VQA is much more diverse than other KB-VQA datasets such as ViQuAE (Lerner et al., 2022), InfoSeek (Chen et al., 2023), and Encyclopedic-VQA (Mensink et al., 2023). Unlike these existing resources, SK-VQA utilizes images from a larger variety of sources because our generation methodology does not rely on linking images to Wikipedia pages in order to obtain context documents. To demonstrate the utility of SK-VQA, we first conduct zero-shot experiments on six state-of-the-art MLLMs, showing that it is a challenging benchmark for these powerful models. We then fine-tune MLLMs on our dataset and compare their performance to models fine-tuned on existing KB-VQA datasets. Our experiments show that SK-VQA enhances the generalization capabilities of MLLMs, whereas other datasets result in poor generalization performance. We attribute this improved generalization capacity to the diversity of our dataset. To summarize, our contribution are threefold:

- We create the largest and most diverse multimodal dataset for KB-VQA to-date. Unlike existing datasets which are limited to images that can be linked to Wikipedia passages, our dataset encompasses a broader range of images, features a more diverse set of question types, and possesses richer linguistic style. Additionally, human evaluations affirm the correctness of the generated QA pairs and the factuality of the generated context documents. Our

dataset[1] and its generation code[2] are publicly available.

- We perform zero-shot evaluations and fine-tuning of several state-of-the-art MLLMs on both our dataset and existing datasets. The zero-shot results indicate that our dataset is more challenging compared to others, demonstrating its complexity despite being synthetically produced. In addition, the fine-tuning results demonstrate that our dataset consistently improves the out-of-domain performance across model sizes on multiple datasets.
- We conduct detailed ablation experiments to evaluate the performance of fine-tuned models across different image types and test their performance within a real-world RAG environment. The experimental results indicate that our dataset effectively enhances the model's out-of-domain performance.

## 2. Related Work

**Synthetic Data Generation** Synthetic data has grown in popularity lately as an effective strategy for data augmentation, particularly in the multimodal domain where data is often more scarce. Advances in text-to-image diffusion models (Nichol et al., 2021; Rombach et al., 2021; Saharia et al., 2022; Ramesh et al., 2022) have enabled the generation of synthetic data for a variety of use cases such as image classification (He et al., 2022; Trabucco et al., 2023; Vendrow et al., 2023) and image-text counterfactuals (Le et al., 2024; Howard et al., 2024). In the domain of NLP, augmenting prompts with LLM-generated context documents has been demonstrated to be competitive with retrieving real text documents for context augmentation in RAG systems (Yu et al., 2022). Synthetic data has also been shown to be useful for training text embedding models for retrieval (Wang et al., 2023). To the best of our knowledge, our work is the first to explore the use of fully synthetic datasets to adapt MLLMs for context-augmented generation.

Despite its demonstrated benefits, several risks have been noted in utilizing synthetic data for training. Shumailov et al. (2023) showed that training language models on data that is contaminated with increasing amounts of model-generated content leads to model collapse, while Gerstgrasser et al. (2024) found that accumulating model-generated content without replacing original content can avoid this. In the context of training vision-language models, synthetic image data has been shown to scale similarly in effectiveness of CLIP (Radford et al., 2021) training as real images, while significantly under performing real data in training supervised image classifiers (Fan et al., 2023). Improving out-of-domain generalization by training on synthetic data has also been shown to be sensitive to the ratio of real and synthetic data (Howard et al., 2022; Le et al., 2024).

---

[1] Our dataset is available via Hugging Face Hub
[2] Our code is available via GitHub

**Knowledge-Based Visual Question Answering Datasets**
Marino et al. (2019) introduced OK-VQA, a KB-VQA dataset of 14k crowdsourced questions for COCO images which are designed to require external knowledge to answer, but are not associated with ground truth context documents. Lerner et al. (2022) introduced the ViQuAE dataset, which consists of 3.7k questions about named entities paired with images and text articles from Wikipedia. Chen et al. (2023) found that many questions in OK-VQA and ViQuAE can be answered without external knowledge; motivated by this finding, they introduced the InfoSeek dataset containing over 1.3 million information-seeking questions paired with images from existing image classification and retrieval datasets which have been grounded to Wikipedia articles. Although they curate a smaller set of 8.9k human-written questions for testing, the vast majority of InfoSeek is automatically constructed by populating human-authored templates from Wikidata triples.

Encyclopedic VQA (Mensink et al., 2023) is another recently-proposed KB-VQA dataset consisting of 221k unique QA pairs which are each associated with up to 5 images from the iNaturalist (Van Horn et al., 2021) and Google Landmarks (Weyand et al., 2020) datasets. They utilize the WIT dataset (Srinivasan et al., 2021) to link images with Wikipedia text documents and employ templates along with a question generation model to automatically construct 1 million question-answer pairs. SnapNTell (Qiu et al., 2024) also contains KB-VQA questions requiring entity-specific external knowledge to answer, but contains fewer QA pairs (75.6k) and was not publicly available at the time of writing. Other knowledge-intensive VQA datasets have been proposed for more specific domains of multimodal documents, including technical engineering requirements (Doris et al., 2024) and scientific journal articles (Ding et al., 2024).

**Multimodal RAG systems** In the domain of KB-VQA, augmenting transformer-based generators with retrieved multimodal documents has been shown to be effective in architectures such as RA-CM3 (Yasunaga et al., 2022), MuRAG (Chen et al., 2022), and REVEAL (Hu et al., 2023). More recently, LLMs augmented with vision encoders such as LLaVA (Liu et al., 2024b) and GPT-4 (Achiam et al., 2023) have demonstrated state-of-the-art performance on image-to-text generation tasks, motivating the investigation of retrieval-based context augmentation for such models. Re-ViLM (Yang et al., 2023) augments Flamingo with retrieved documents, while Wiki-LLaVA (Caffagni et al., 2024) augments LLaVA model with Wikipedia documents.

Wei et al. (2023) proposed UniIR for multimodal retrieval, utilizing score-level and feature-level fusion approaches with pre-trained CLIP (Radford et al., 2021) and BLIP (Li et al., 2022) models. Sharifymoghaddam et al. (2024) showed that UniIR can improve the performance of large

multimodal language models on image captioning and image generation tasks. UniMur (Wang et al., 2024b) embeds multimodal inputs and retrieves multimodal outputs via frozen LLMs. Our work differs from these studies in that we focus on how to adapt MLLMs for context-augmented generation in a RAG system rather than training the retriever.

## 3. Methodology

### 3.1. Dataset generation

Motivated by recent advances in MLLMs, we use a fully automated approach to generate synthetic context documents and question-answer (QA) pairs for a given image with GPT-4. This provides several distinct advantages. First, the powerful language abilities of GPT-4 allow us to acquire more natural and diverse questions than previous datasets which rely on templated construction of question-answer pairs. Second, generating context documents enables the use of a much broader range of images than previous datasets, where images are typically restricted only to those that can be linked to Wikipedia passages.

Given an input image, we prompt GPT-4 to generate a context document[3] related to the image and QA pairs which require reasoning over both the image and the context document. The complete prompt we use for this purpose is provided in Figure 3 (see Figure 7 of Appendix H for additional discussion). Importantly, we generate both the context document and QA pairs in a single inference step. In doing so, the generation of the context is conditioned on the task of producing questions that require both the image and the context. This helps ensure that the context associated with each image is suitable for the creation of the style of QA pairs we seek, which is not necessarily the case when context documents are acquired automatically from existing sources such as Wikipedia. Following generation, we parse the output of GPT-4 to extract the context document and QA pairs (see Appendix H for details). We then apply two stages of filtering to create separate filtered subsets.

### 3.2. Image Reference (IR) filtering

In the manual evaluation of generated context documents, we found that GPT-4 sometimes directly references the input image that was provided. For example, the context documents may include references such as "In the image, ..." or "As shown in the picture, ...". In such cases, the information contained in the context document is more similar to an extended caption or image description than a knowledge-intensive document. While this may not necessarily be detrimental to the training of multimodal RAG systems, it is unlikely in practice for RAG systems to require the retrieval

---

[3]See Appendix O for a discussion of hallucination impact

```
Write a Wikipedia article related to
this image without directly referring to
 the image. Then write question answer
pairs. The question answer pairs should
satisfy the following criteria.

1: The question should refer to the
image.
2: The question should avoid mentioning
the name of the object in the image.
3: The question should be answered by
reasoning over the Wikipedia article.
4: The question should sound natural and
 concise.
5: The answer should be extracted from
the Wikipedia article.
6: The answer should not be any objects
in the image.
7: The answer should be a single word or
 phrase and list all correct answers
separated by commas.
8: The answer should not contain 'and',
'or', rather you can split them into
multiple answers.
```

*Figure 3.* Our prompt for generating synthetic data.

of image-specific context documents. We therefore create a filtered subset of our dataset which excludes these cases by identifying the presence of the words `picture`, `photo`, `image`, or `painting` in the generated context document. We refer to this subset as SK-VQA$_{IR}$.

### 3.3. Context Answer Presence (CAP) filtering

In existing datasets for KB-VQA, it is common for the answer to be explicitly stated in the associated context document. This is not necessarily required in order for a QA pair to be valid since the answer could sometimes be inferred indirectly rather than being explicitly stated in the context. Nevertheless, the presence of the answer in the context document provides an indication that the question can indeed be answered from information contained in it. Therefore, we create an additional filtered subset of our dataset which only contains QA pairs where (1) at least one of the answer candidates is contained in the context document, and (2) the context does not directly reference the image (as described previously). We refer to this subset as SK-VQA$_{IR+CAP}$.

## 4. Dataset Analysis

### 4.1. Dataset Composition

In order to acquire synthetic data which spans a broad range of different domains, we utilize images from multiple sources during generation. These sources include LAION-400m, Wikipedia images contained in the WIT dataset, and synthetically generated images from COCO-

*Table 1.* Total number of QA pairs in our dataset by image and context source, computed separately for each filtered subset.

| Image source | Context source | SK-VQA | SK-VQA$_{IR}$ | SK-VQA$_{IR+CAP}$ |
|---|---|---|---|---|
| LAION | GPT-4 | 908,116 | 584,126 | 371,936 |
| Wikipedia | GPT-4 | 702,332 | 585,768 | 354,244 |
| Wikipedia | Wikipedia | 181,554 | 167,352 | 137,160 |
| COCO-CFs | GPT-4 | 214,487 | 193,226 | 121,284 |
| | | 2,006,489 | 1,530,472 | 984,624 |

*Table 2.* Comparison of question (Q) diversity in KB-VQA datasets (* values are previously reported by Lerner et al. (2024)).

| Dataset | Total Qs | Unique Qs | Unique POS | Vocab Size | Length |
|---|---|---|---|---|---|
| ViQuAE* | 3,700 | 3,562 | 2,759 | 4,700 | 12.4 |
| InfoSeek* | 1,356,000 | 1,498 | 267 | 725 | 8.9 |
| Enc-VQA* | 1,036,000 | 175,000 | 91,945 | 40,787 | 11.6 |
| **SK-VQA** | **2,006,489** | **1,928,336** | **926,817** | **138,372** | **12.7** |

Counterfactuals (COCO-CFs). We use the entirety of COCO-Counterfactuals along with a sub-sample of images from LAION and Wikipedia to generate context documents with QA pairs using our prompt. We also generate only QA pairs for a sub-sample of Wikipedia images paired with Wikipedia context documents, which enables us to compare the effect of using real context documents to synthetically generated contexts (see Appendix H for details).

Table 1 provides a breakdown of the total number of QA pairs in our dataset by image and context source. Our full SK-VQA dataset contains over 2 million QA pairs, making it the largest KB-VQA dataset created to-date. Of these 2 million QA pairs, 45% are associated with images sourced from LAION, 44% are associated with Wikipedia images, and the remainder are paired with synthetic images from COCO-Counterfactuals. The SK-VQA$_{IR}$ subset contains 24% less QA pairs than the full SK-VQA dataset, while the SK-VQA$_{IR+CAP}$ subset contains approximately half the number of QA pairs as the full SK-VQA dataset.

Our full dataset contains 290,266 unique image-context pairs, which each have 7 QA pairs on average. GPT-4 generated context documents are associated with 7.1 QA pairs on average, whereas Wiki context documents are only associated with 5.7 QA pairs. This indicates that having GPT-4 generate both the context document and QA pairs simultaneously enables the acquisition of more QA pairs, which could be attributable to how the generation of the context document is conditioned on the subsequent task of producing QA pairs.

### 4.2. Question Diversity

Table 2 provides statistics on the diversity of questions in our dataset and other existing KB-VQA datasets. In addition

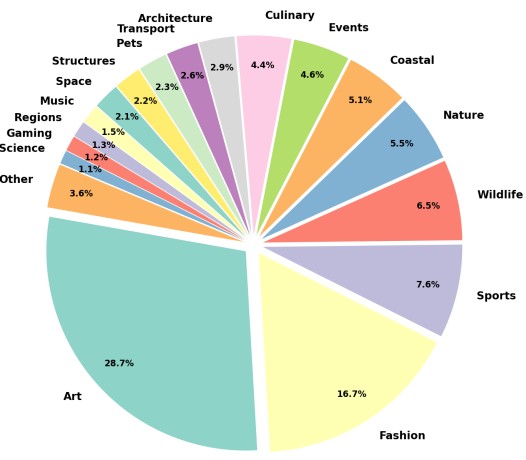

*Figure 4.* Topic distribution for SK-VQA context documents.

to having nearly 50% more questions then the next-largest dataset, our dataset also exhibits significantly greater question diversity. The only two existing datasets of comparable size, InfoSeek and Encyclopedic-VQA (Enc-VQA), contain significantly fewer unique questions; less than 1% of questions in InfoSeek are unique, while fewer than 17% are unique in Enc-VQA. In contrast, over 96% of the questions in SK-VQA are unique, which corresponds to 11x more unique questions than Enc-VQA. Questions in SK-VQA also exhibit a greater number of unique POS sequences, total vocabulary size, and mean word length. This points to the value of leveraging powerful MLLMs for synthetic data generation over simpler techniques (e.g., templates).

### 4.3. Knowledge Classification

We apply an unsupervised topic modeling technique to categorize the knowledge contained in our dataset's context documents (see Appendix J for details). Figure 4 depicts the distribution of major topic categories identified in this analysis. Whereas previous KB-VQA datasets have focused primarily on entity-specific knowledge, our dataset spans a broader range of topics such as art, fashion, sports, events, and music. This demonstrates the diversity of external knowledge required by questions in our dataset and suggests that it can serve as a complementary resource to existing datasets which focus on entity-specific knowledge.

### 4.4. Human Evaluation

#### 4.4.1. QUESTION ANSWER PAIRS QUALITY

We randomly sample 100 QA pairs from our dataset for human labeling by three of the authors of this work, ensuring that the 100 QA pairs are equally distributed across the four

*Table 3.* Human performance on different subsets of 100 sampled QA pairs from SK-VQA, calculated using semantic evaluation.

|                  | Mean | Standard Deviation |
|------------------|------|--------------------|
| SK-VQA           | 0.77 | 0.02               |
| SK-VQA$_{IR}$    | 0.77 | 0.02               |
| SK-VQA$_{IR+CAP}$| 0.87 | 0.03               |

image & context source types shown in Table 1. For each QA pair, the annotators were presented only with the image, context document, and question. They were then instructed to write an answer to the question, and optionally note any deficiencies that were evident.

Table 3 provides the mean and standard deviation of annotator accuracy, calculated using the Enc-VQA semantic evaluation method (see Section 5.2 for details). The overall mean accuracy of the three annotators was 77% for the 100 QA pairs sampled from SK-VQA and the subset of those which belong to SK-VQA$_{IR}$. For the subset of annotated QA pairs which belong to SK-VQA$_{IR+CAP}$, the mean human accuracy increases to 87%, which is consistent with previously reported human accuracy for other VQA datasets (Hudson & Manning, 2019; Sheng et al., 2021). The relatively low standard deviation indicates annotators performed similarly.

To understand potential failure cases in our dataset, we categorized common annotator comments by identifying those for which at least two annotators recorded the same category of issue for a question. The most common issue reported by annotators were cases in which the question could be answered solely using the context document, assuming that the context document was provided at inference time. While this concern was noted for 9% of evaluated QA pairs, these examples may still require multimodal reasoning in a broader RAG system in which the question and image are necessary to retrieve the relevant context document. A small number of examples (5%) were identified as being answerable solely by looking at the image, while 1 question was noted as having insufficient information in the context and image to answer (see Figure 12 for examples).

### 4.4.2. FACTUALITY EVALUATION

Factual accuracy is not a primary concern for our synthetic data, as its main purpose is to train MLLMs to effectively utilize long contexts for VQA. Nevertheless, we conduct analyses to validate factual accuracy. Initially, we implement the automated fact-checking method SAFE (Wei et al., 2024). However, we find it ineffective for our dataset because it misclassified image-descriptive sentences as unsupported due to its inability to process multimodal inputs.

Therefore, we conduct a human evaluation to assess factuality. Specifically, we ask a native speaker to fact-check

50 QA pairs and supporting evidence in context documents using online sources. 86% are verified as factual, 4% are non-factual, 2% are partially factual, and 8% can not be determined due to a lack of available information.

### 4.5. Automatic Dataset Quality Evaluations

We conduct comprehensive automatic quality assessments of our generated dataset through multiple methods.

### 4.5.1. RULE-BASED AND NLP TOOL EVALUATION

We assess the quality of our generated context documents through grammatical evaluation. Specifically, we use the widely-used LanguageTool [4] on a random sample of 10K context documents. Out of 436k characters, only 6.69 % are flagged for grammatical issues. A manual review of 50 flagged sentences showed that 80% are minor issues related to spelling, style, or punctuation. This reinforces our confidence that the generated text maintains high grammatical accuracy.

Another potential concern with synthetically generated data is the presence of biases, which can be reflective of those possessed by the models used for generation. We use state-of-the-art bias[5] and toxicity[6] detection models to examine all generated documents for potential bias or toxic content. The detection results show no bias or toxicity in our dataset.

### 4.5.2. LLM-AS-JUDGE EVALUATION

To evaluate the quality of our dataset at scale, we employ an LLM-as-judge methodology using GPT-4o (OpenAI, 2024), a different model from the one used for data synthesis. This automated evaluation targets four key dimensions of QA quality: (1) factuality of the image description, (2) relevance of the question to the image, (3) answerability of the question given the context, and (4) factual correctness of the answer.

For image-description factuality, we prompt GPT-4o to assess how accurately the textual description reflects the image content on a scale from 0 (completely inaccurate) to 5 (fully accurate). For QA pair quality, we evaluate three aspects: relevance of the question to the description (0–5 scale), answerability based on the description (Yes/No), and factual correctness of the answer with respect to the description (Yes/No). The detailed prompts are provided in Tables 12 and 13.

Using this automated framework, we conduct a large-scale evaluation of our synthetic dataset. The results demonstrate high quality across all dimensions:

---

[4]LanguageTool via jxmorris12
[5]NLP Bias Detector via vector institute
[6]RoBERTa Detector via Facebook

- **Factuality of Descriptions:** Average score of 4.6/5, with 87.5% receiving perfect scores, indicating strong alignment between generated descriptions and visual content.

- **Question Relevance:** Average score of 4.9/5, with 92.0% rated at the highest level, demonstrating well-aligned questions with image context.

- **Answerability:** 99.6% of questions are clearly answerable based on the provided context, suggesting high internal coherence.

- **Answer Correctness:** 90.7% of answers are factually correct with respect to the description, indicating reliable generated answers.

Together with our human evaluation, these automated assessments provide scalable and interpretable validation of our dataset quality across multiple dimensions, confirming the high quality of SK-VQA for training and evaluating MLLMs on context-augmented VQA tasks.

### 4.6. Comparison with Existing Datasets

We conduct an analysis to compare our dataset to two types of multimodal datasets w.r.t. comprehensiveness and data quality: VQA datasets and those generated by LLMs. A detailed summary of our findings is provided in Table 11 of the Appendix. Existing VQA datasets are typically created using pre-defined templates or generated manually, which limits their diversity and scalability. On the other hand, previous datasets generated by LLMs are often not verified by humans. In contrast, SK-VQA is both comprehensive in size and has been validated to possess high quality through human evaluations which exceed those employed for other LLM-generated datasets.

## 5. Experiments

### 5.1. Experimental Setup

We conduct zero-shot and fine-tuning experiments on MLLMs using both our dataset and existing KB-VQA datasets. For the zero-shot experiments, we test the following popular MLLMs: PaLIGemma-3B (Beyer et al., 2024), LLaVA-v1.5-7B (Liu et al., 2024b), LLaVA-1.6-7B/34B (Liu et al., 2024a), Qwen-VL-7B (Bai et al., 2023), and Idefics2-8B (Laurençon et al., 2024). For the fine-tuning experiments, we utilize LLaVA-v1.5-7B and PaLI-Gemma-3B. To demonstrate the effectiveness of our synthetic data, we train models using various subsets of our dataset generated from different sources, as described previously in 3.1. Additionally, we train two baseline models on existing KB-VQA datasets: InfoSeek and Enc-VQA. For InfoSeek, we use a 140K subset of the training data processed by Wei

et al. (2023), where only external textual knowledge is required for the given questions and images (denoted as as task 6 by Wei et al. (2023)). We use the original Enc-VQA training set, but since each question can be paired with multiple images, we select only the first image from the original annotations for the training set, which results in approximately 220K training samples. For a fair comparison, we down-sample our dataset subsets to 200K samples each. For the real Wikipedia context from the WIT dataset, we use the paragraph-level context associated with each image. See Appendix F and G for additional details.

### 5.2. Evaluation Datasets and Metrics

We use three existing KB-VQA datasets for evaluation: InfoSeek, Enc-VQA, and ViQuAE. Additionally, we use 10,744 samples from SK-VQA$_{IR}$ associated with images from LAION for model evaluation. For InfoSeek, we use a subset of its validation set processed by Wei et al. (2023), which includes 11,323 samples where only external textual knowledge is required for the given questions and images. For Enc-VQA, we use its official test set, which contains 5,750 samples. Due to the small size of the ViQuAE test set, we combine the train, validation, and test sets to create a larger testing set of 3,625 samples. We use the official semantic evaluation method for Enc-VQA, BEM (Bulian et al., 2022). For other datasets, we use exact string matching.

### 5.3. Fine-tuning Results

We evaluate fine-tuned MLLMs on their ability to generalize to other datasets (i.e., out-of-domain performance). Figure 5 shows that for LLaVA-7B, fine-tuning with the InfoSeek dataset enhances performance on the SK-VQA test set but does not yield improvements on Enc-VQA or ViQuAE. Similarly, fine-tuning with Enc-VQA fails to surpass baseline performance across all datasets. In contrast, models trained on our SK-VQA dataset achieve significant zero-shot improvements on both InfoSeek and Enc-VQA while outperforming the models trained on the other two datasets on ViQuAE. We also trained LLaVA-13B models, resulting in similar trends (see Appendix A for details).

Fine-tuning PaliGemma-3B using InfoSeek results in performance degradation in 2 out of 3 settings compared to the zero-shot baseline. However, fine-tuning with Enc-VQA consistently improves performance. Again, models trained on our dataset show significant performance improvements in all 9 cases and achieve the best out-of-domain performance. Overall, these results indicate that fine-tuning MLLMs with SK-VQA effectively improves out-of-domain performance in most cases. Even when there is no improvement, SK-VQA does not result in performance degradation as with fine-tuning on other datasets. The in-domain performance, as expected, shows that the model performs best on

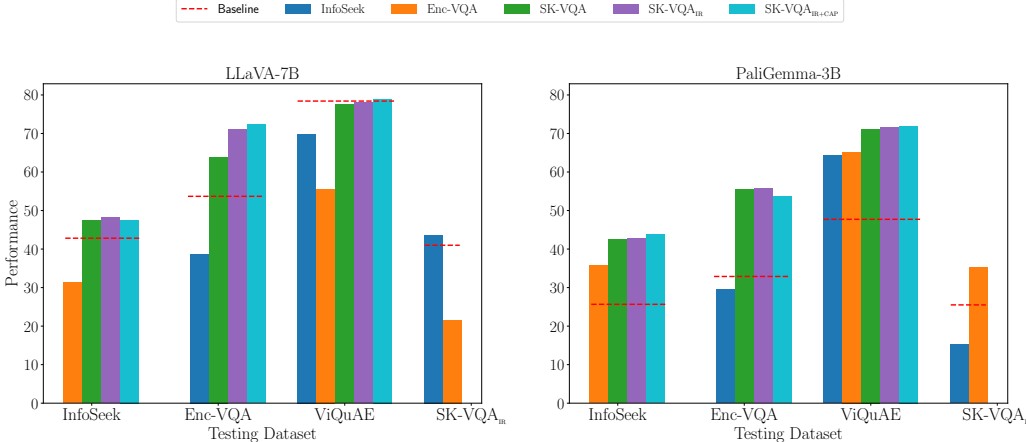

*Figure 5.* Performance of models trained on different KB-VQA datasets (indicated by bar colors) and tested on various datasets (x-axis labels). Red dashed lines indicate baseline performance of the MLLM without training.

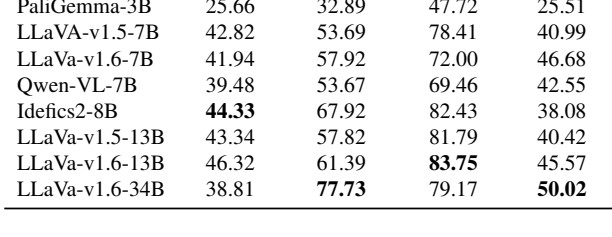

*Table 4.* Zeroshot evaluation of SOTA MLLMs on existing KB-VQA and our datasets.

| Model | Infoseek | Enc-VQA | ViQuAE | SK-VQA |
|---|---|---|---|---|
| PaliGemma-3B | 25.66 | 32.89 | 47.72 | 25.51 |
| LLaVA-v1.5-7B | 42.82 | 53.69 | 78.41 | 40.99 |
| LLaVa-v1.6-7B | 41.94 | 57.92 | 72.00 | 46.68 |
| Qwen-VL-7B | 39.48 | 53.67 | 69.46 | 42.55 |
| Idefics2-8B | **44.33** | 67.92 | 82.43 | 38.08 |
| LLaVa-v1.5-13B | 43.34 | 57.82 | 81.79 | 40.42 |
| LLaVa-v1.6-13B | 46.32 | 61.39 | **83.75** | 45.57 |
| LLaVa-v1.6-34B | 38.81 | **77.73** | 79.17 | **50.02** |

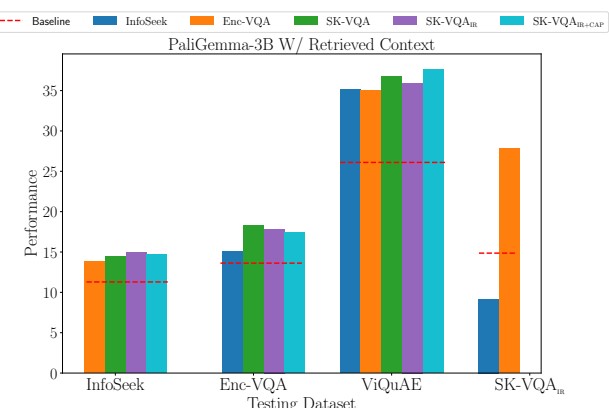

*Figure 6.* Generalization Performance of fine-tuned PaliGemma in RAG setup. Our models achieve the best generalization.

the data it was fine-tuned on (see Appendix Table 7).

## 5.4. Zero-shot Evaluation Results

Table 4 provides the results for zero-shot evaluations. All tested state-of-the-art MLLMs perform better on Enc-VQA and ViQuAE compared to InfoSeek and our SK-VQA dataset. This suggests that SK-VQA and InfoSeek present significant challenges to these models. Unlike with Enc-VQA and ViQuAE, larger models do not always yield better performance on InfoSeek and SK-VQA. This indicates that simply relying on model size may not be sufficient to address the reasoning challenges presented by these datasets. Additional evaluation results on recent state-of-the-art models can be found in Appendix D.

## 5.5. Ablation Studies

**Retrieval Augmented Generation (RAG) Results** In addition to utilizing the gold-label context documents (Tables 4 & 7), we use the CLIP Score Fusion model from Wei et al. (2023) to retrieve knowledge from external text knowledge bases as context to simulate a real RAG setup. This setting

is more challenging because the model needs to distinguish relevant parts of the context from irrelevant information. In this experiment, we focus on using the Paligemma-3B model (see Appendix A for the construction of external knowledge bases). For each question, we retrieve the top 10 most relevant passages. During inference, we combine each of these 10 retrieved passages with the question and perform inference. The final answer is determined by selecting the most frequently occurring answer among these 10 inferences. Results by model are presented in Figure 6. The results show that even when using retrieved contexts, the model trained on our dataset performs strongly both in-domain and out-of-domain. Notably, in out-of-domain performance, it surpasses the baseline zero-shot performance and models trained on the other two datasets in all 9 cases.

**Impact of Generation Source** We explore the performance of models trained on data generated from different

*Table 5.* Performance comparison of LLaVa-v1.5-7B trained on synthetic data derived from different sources (without filtering).

| Image Source | Context Source | Infoseek | Enc-VQA | ViQuAE | Avg. |
|---|---|---|---|---|---|
| LAION | GPT-4 | 44.32 | 65.44 | 79.22 | 62.99 |
| Wiki | GPT4 | 47.00 | 53.98 | 78.58 | 59.85 |
| Wiki | Wiki | 47.75 | **66.67** | 77.95 | 64.12 |
| COCO-CFs | GPT4 | **48.00** | 65.42 | **79.23** | **64.22** |

*Table 6.* Zeroshot evaluation of SOTA MLLMs on existing KB-VQA and our datasets.

| Model | LAION | WiT(GPT-4) | WiT(Wiki) | Coco-CF |
|---|---|---|---|---|
| LLaVA-v1.5-7B | 40.99 | 44.35 | 50.45 | 41.4 |
| LLaVa-v1.6-7B | 46.68 | 48.9 | 54.8 | 46.85 |
| Qwen-VL-7B | 42.55 | 42.45 | 47.6 | 41.6 |
| LLaVa-v1.5-13B | 40.42 | 41.5 | 50.85 | 39.4 |
| LLaVa-v1.6-13B | 45.57 | 46.5 | 56.25 | 43.5 |

images and context sources to understand how each image source contributes to the dataset's effectiveness. Table 5 shows that the best combination involves using images from COCO-CFs and context documents from GPT-4. Notably, this combination even surpasses using images from Wikipedia with their real context, demonstrating that synthetic images paired with generated contexts can be as effective as—or even more effective than—real image-context pairs. This indicates that SK-VQA can offer advantages for fine-tuning MLLMs compared to real data. Additionally, when using GPT-4 generated context documents, we observe that models fine-tuned on data derived from LAION images perform better on Enc-VQA and ViQuAE, whereas models fine-tuned on data derived from Wiki images perform better on InfoSeek. This suggests that different image sources contribute distinct generalization capabilities, and that combining images from diverse sources—including synthetic ones—may be necessary to achieve better generalization across all external datasets, as shown in §5.3.

**Exploration of Using LLaVA-34B for Generation**  We also explored the use of the state-of-the-art open-source model LLaVA-34B as a replacement for GPT-4 in generating synthetic data. We follow the same pipeline mentioned earlier and manually evaluate the generated data. However, the annotators report that 76% of the questions generated by LLaVA-34B are invalid due to one of the three reasons discussed in §4.1. Most of these questions can be answered solely by the context, without requiring the image, such as "What is the purpose of the Great Wall?" or "What is the main purpose of a dining set?". Additionally, we find that the questions generated by LLaVA-34B are much simpler compared to those generated by GPT-4, often not requiring complex reasoning to answer. These findings suggest that maintaining the quality of generated datasets requires using the most advanced MLLM, even if it is proprietary.

## 6. Conclusion

We introduced SK-VQA: a large dataset of 2 million QA pairs over images from multiple different sources. SK-VQA is the largest and most diverse resource of its kind, possessing 11x more unique questions than similar datasets for KB-VQA. Our evaluations of popular MLLMs showed

that our dataset can serve as a more challenging benchmark than existing resources. Additionally, training MLLMs on our dataset leads to greater improvements in out-of-domain generalization than other datasets. These results point to not only the utility of SK-VQA, but also the effectiveness of our approach for acquiring synthetic multimodal data at scale. Opportunities for future work in this direction include leveraging larger amounts of synthetic image data to produce fully-synthetic datasets for other domains and leveraging SK-VQA for training multimodal retrieval models to aid in the development of RAG systems.

## Impact Statement

This paper presents work whose goal is to advance the field of multimodal machine learning through the creation of a large-scale synthetic dataset for knowledge-based visual question answering. While our primary contribution is to improve the training and evaluation of multimodal language models, we acknowledge several broader implications of our work.

The use of synthetic data generation at scale presents both opportunities and risks. On the positive side, our approach democratizes access to high-quality training data, potentially enabling researchers with limited resources to develop competitive multimodal systems. However, as discussed in our limitations (Appendix O), the synthetic nature of our dataset means it may contain inaccuracies or reflect biases present in GPT-4, despite content filtering measures. We encourage users of our dataset to validate model performance thoroughly before deployment.

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

*Table 7.* Performance of Generator fine-tuned on ~200K from different datasets. * denotes the in-domain results. Encyclopedia evaluated on semantic metric, others are Exact Matching scores.

| | Training Data | Infoseek | Enc-VQA | ViQuAE | SK-VQA$_{IR}$ |
|---|---|---|---|---|---|
| **PaliGemma-3B** | - | 25.66 | 32.89 | 47.72 | 25.51 |
| | Infoseek | 66.63* | 29.58 | 64.22 | 15.27 |
| | Enc-VQA | 35.78 | 83.30* | 65.19 | **35.29** |
| | SK-VQA | 42.50 | 55.53 | 71.12 | 65.02* |
| | SK-VQA$_{IR}$ | 42.69 | **55.67** | 71.48 | 65.26* |
| | SK-VQA$_{IR+CAP}$ | **43.72** | 53.72 | **71.97** | 64.69* |
| **LLaVA-7B** | - | 42.82 | 53.69 | 78.41 | 40.99 |
| | Infoseek | 68.68* | 38.68 | 69.90 | **43.69** |
| | Enc-VQA | 31.39 | 88.75* | 55.59 | 21.60 |
| | SK-VQA | 47.35 | 63.89 | 77.60 | 68.30* |
| | SK-VQA$_{IR}$ | **48.11** | 70.99 | 78.04 | 68.35* |
| | SK-VQA$_{IR+CAP}$ | 47.55 | **72.33** | **78.95** | 68.61* |
| **LLaVA-1.6-13B** | - | 46.32 | 61.39 | **83.75** | 45.57 |
| | Infoseek | 74.48* | 54.59 | 79.56 | **52.98** |
| | Enc-VQA | 40.36 | 90.59* | 72.88 | 29.71 |
| | SK-VQA | 47.58 | 66.84 | 82.04 | 70.27* |
| | SK-VQA$_{IR}$ | **50.28** | 66.58 | 81.02 | 70.23* |
| | SK-VQA$_{IR+CAP}$ | 48.77 | **67.35** | 81.79 | 70.79* |

## A. Additional Fine-tuning Experimental Results

**LLaVA-1.6-13B Fine-tuned Results**   Table 7 showcases the performance of fine-tuning LLaVA-1.6-13B on different datasets. The models trained on our datasets achieve the best generalization performance compared to the models trained on other two existing datasets.

**In-domain performance**   Table 7 provides additional in-domain performance results of fine-tuned models from the experiments discussed in §5.3. The results indicate that models trained on specific datasets perform best on their corresponding test sets. However, as noted in §5.3, good in-domain performance for models trained on InfoSeek and Enc-VQA does not guarantee good out-of-domain performance. In contrast, our models, trained on our dataset, achieve strong performance both in-domain and out-of-domain.

**Retrieval Augmented Generation External Knowledge Bases Construction**   For constructing external knowledge bases, we use the InfoSeek dataset knowledge base processed by Wei et al. (2023), which includes 611,651 passages. For the other three datasets, Enc-VQA, InfoSeek, and ViQuAE, we create synthetic external knowledge bases by merging the corresponding gold passages for each test set question. The sizes of the knowledge bases for Enc-VQA, ViQuAE, and our synthetic dataset are 3,859, 71,985, and 1,514 passages, respectively.

## B. Ablation Study on Impact of Filtering Techniques

To further explore the impact of different filtering methods and their interaction with image sources, we fix the image source for data generation and compare the performance of models trained on data filtered by various methods across three out-of-domain datasets, as shown in Table 8. We sequentially apply IR and CAP filtering on SK-VQA. For data from LAION, SK-VQA$_{IR}$ retains 64% of the original data, while SK-VQA$_{IR+CAP}$ retains 40%. For data from Wikipedia, SK-VQA$_{IR}$ retains 83%, and SK-VQA$_{IR+CAP}$ retains 50%.

The results reveal how filtering methods interact differently with various image sources. With LAION as the image source, SK-VQA$_{IR+CAP}$ achieves the best average performance across the three datasets, though SK-VQA outperforms SK-VQA$_{IR}$. For data from Wikipedia, SK-VQA$_{IR}$ outperforms both SK-VQA$_{IR+CAP}$ and SK-VQA overall, while certain datasets (InfoSeek) benefit most from the full SK-VQA dataset. This demonstrates that LAION and Wiki sources exhibit

*Table 8.* Impact of filtering techniques by image source. All results utilize context documents from GPT-4.

| Training Data | Image Source | Infoseek | Enc-VQA | ViQuAE | Avg. |
|---|---|---|---|---|---|
| SK-VQA | LAION | 44.32 | 65.44 | **79.22** | 62.99 |
| SK-VQA$_{IR}$ | LAION | 44.43 | 63.08 | 75.50 | 61.00 |
| SK-VQA$_{IR+CAP}$ | LAION | **45.85** | **69.88** | 78.18 | **64.64** |
| SK-VQA | Wiki | **47.00** | 53.98 | 78.58 | 59.85 |
| SK-VQA$_{IR}$ | Wiki | 45.99 | **67.36** | 79.37 | **64.24** |
| SK-VQA$_{IR+CAP}$ | Wiki | 46.48 | 64.55 | **79.83** | 63.62 |

distinct generalization patterns when combined with different filtering strategies, suggesting that specific filtering methods may improve performance for certain domains or datasets, while the full unfiltered SK-VQA dataset might be more valuable for others. These findings highlight the versatility of our multi-source approach and filtering techniques. The various filtered subsets of our dataset can be treated as a hyperparameter to find the best performance for specific tasks. We also note that a significant benefit of filtering is the ability to achieve similar or better performance with significantly fewer samples, which holds true for both filtering methods across all datasets.

## C. Difficulty Analysis Across Different Image Sources

To better understand the role of different image sources in our dataset, we evaluate the performance of state-of-the-art multimodal LLMs on different subsets of SK-VQA, divided based on the source and type of image content. Specifically, we test LLaVA-v1.5 (7B and 13B), LLaVA-v1.6 (7B and 13B), and Qwen-VL (7B) (Bai et al., 2023) across four distinct subsets: WiT (Wikipedia context), WiT (GPT-4 generated context), LAION, and COCO-CF.

*Table 9.* Model performance across different image source subsets of SK-VQA. Results show consistent difficulty ordering across all models.

| Model | LAION | WiT (GPT-4) | WiT (Wiki) | COCO-CF |
|---|---|---|---|---|
| LLaVA-v1.5-7B | 40.99 | 44.35 | 50.45 | 41.40 |
| LLaVA-v1.6-7B | 46.68 | 48.90 | 54.80 | 46.85 |
| Qwen-VL-7B | 42.55 | 42.45 | 47.60 | 41.60 |
| LLaVA-v1.5-13B | 40.42 | 41.50 | 50.85 | 39.40 |
| LLaVA-v1.6-13B | 45.57 | 46.50 | 56.25 | 43.50 |
| Average | 43.24 | 44.74 | 51.99 | 42.55 |

The results in Table 9 demonstrate that each subset presents a distinct level of difficulty. We observe a consistent pattern across all models, with difficulty increasing in the following order: WiT (Wikipedia context), WiT (GPT-4 generated context), LAION, and COCO-CF.

We hypothesize that the WiT (Wiki) subset achieves the highest performance because LLMs are likely trained on substantial amounts of Wikipedia content, making this subset more familiar and easier to process. In contrast, the COCO-CF subset, which includes counterfactual images paired with GPT-4 generated content, presents the highest degree of difficulty. These counterfactual image-content pairs are largely out-of-distribution relative to the training data of existing models, resulting in consistently lower performance.

These findings highlight the diversity of our dataset and underscore the importance of incorporating varied content sources in the construction of SK-VQA. While many existing knowledge-based VQA datasets predominantly rely on Wikipedia-based images, our results demonstrate that including diverse sources—particularly those beyond Wikipedia—creates a more challenging and comprehensive benchmark for evaluating multimodal LLMs.

## D. Evaluation on Additional State-of-the-Art Multimodal LLMs

To explore the generalization of our benchmark across a broader range of architectures and model scales, we evaluate additional state-of-the-art multimodal LLMs on SK-VQA. Specifically, we test GPT-4o (OpenAI, 2024), two recently

released model families: Qwen-2.5-VL (3B, 7B, 32B, and 72B parameters) (Team, 2024) and Ovis (1B to 34B parameters) (Lu et al., 2024), which represent some of the most advanced multimodal models currently available. We use the same test set as described in Section 5 to ensure fair comparison with previously reported results.

*Table 10.* Performance of additional state-of-the-art multimodal LLMs on SK-VQA compared to ViQuAE. Results demonstrate that SK-VQA provides a consistent and challenging benchmark across diverse model architectures and scales.

| Model | SK-VQA | ViQuAE |
|---|---|---|
| GPT-4o | 58.90 | – |
| Qwen-2.5-VL-3B | 53.74 | – |
| Qwen-2.5-VL-7B | 49.26 | – |
| Qwen-2.5-VL-32B | 52.08 | – |
| Qwen-2.5-VL-72B | 49.09 | – |
| Ovis-1B | 32.25 | 39.50 |
| Ovis-2B | 44.54 | 67.09 |
| Ovis-4B | 50.55 | 49.38 |
| Ovis-8B | 50.36 | 57.96 |
| Ovis-16B | 52.36 | 72.77 |
| Ovis-34B | 55.20 | 67.03 |

The results in Table 10 reveal several important findings. First, even the most advanced models achieve moderate performance on SK-VQA, with GPT-4o reaching 58.90% accuracy and the best-performing open model (Ovis-34B) achieving 55.20%. This indicates that SK-VQA captures complex multimodal reasoning challenges that remain difficult for current state-of-the-art models.

Second, comparing the Ovis models' performance on SK-VQA versus ViQuAE demonstrates that our dataset provides a more consistent evaluation across model scales. While Ovis models show highly variable performance on ViQuAE (ranging from 39.50% to 72.77%), their performance on SK-VQA remains within a narrower range (32.25% to 55.20%). This consistency suggests that SK-VQA avoids the evaluation instabilities observed in existing benchmarks.

## E. LLM Evaluation

Previous studies have shown that a significant proportion of questions in OK-VQA and ViQuAE can be answered by an LLM when prompted with only the question (Chen et al., 2023). To investigate whether this is the case for our dataset, we generate answers from LLaMA-3-70b-Instruct for all 2 million QA pairs in our dataset using the following prompt:

```
Write a single word or phrase which
answers the question.
Question: [QUESTION]
```

where [QUESTION] is populated with questions from our dataset at query time. We found that the exact match accuracy of LLaMA-3-70b is only 9.92% for our dataset, indicating that the vast majority of questions do indeed require reaosning over the associated images and context documents.

## F. MLLMs Zero-shot Prompts

For all MLLMs in 4, we use the following text prompt when conducting zero-shot prompting, in addition to each model's specific image token:

```
Context {context} Based on the context,
{question} answer the question using
a single word or phrase.
```

```
 1  Write a Wikipedia article related to this image without directly referring to the image
    . Then write question answer pairs. The question answer pairs should satisfy the
    following criteria.
 2
 3  1: Guide the model to generate questions using image information.
 4  2: Avoid questions that can be answered without looking at the image.
 5  3: Guide the model to generate questions using external context rather than simple
    visual information from the image.
 6  4: Since GPT-4 tends to generate unnecessarily lengthy questions that do not sound
    natural, this condition helps to prevent such questions.
 7  5: Guide the model to utilize the context and also make answer evaluation
    straightforward.
 8  6: GPT-4 tends to ask questions where the answer is an object in the image. For such
    questions, context is not needed, which is not of our interest.
 9  7: We can split multiple correct answers into a list to make the evaluation easier.
10  8: This condition is also for making the evaluation easier.
```

*Figure 7.* Explanation for each prompt condition. The numbered explanations correspond to the numbered conditions in our prompt (Figure 3)

## G. MLLMs Fine-tuning Hyperparameters

We use the official codebase[7] from LLaVA-1.5 to fine-tune the llava-v1.5-7b model[8] and the Trainer from Huggingface Transformers library[9] to fine-tune the paligemma-3b-mix-224 model [10]. For the llava-v1.5-7b model, we use a batch size of 16 and a learning rate of 2e-5, training the model for one epoch using bfloat16. Similarly, for the paligemma-3b-mix-224 model, we use a batch size of 64 and a learning rate of 2e-5, also training for one epoch using bfloat16. These are default hyperparamter values which were not tuned as part of our expieriments. The inputs to the models are a combination of the question, image, and context, and the outputs are the answers to the questions.

## H. Additional details of GPT-4 generation

**Prompts**    Figure 7 provides an explanation of the motivation for each condition which we include in our prompt. The numbered explanations correspond to the numbered conditions in our prompt which are shown in Figure 3. The instruction and conditions in our prompt were derived through manual prompt engineering, where different prompts were tested and iteratively updated in response to issues that were identified in the synthetic data produced by GPT-4.

As discussed in Section 4.1, we also generated only QA pairs for a sub-sample of Wikipedia image-context pairs sourced from the WIT dataset. Figure 8 provides the prompt that we used with GPT-4 for this generation setting. In this prompt, [CONTEXT] is a placeholder where the actual Wikipedia context document is inserted at inference time. Other conditions in this prompt are identical to those in our main prompt specified in Figure 3.

**Output parsing**    Here we describe the process of extracting context, question, and answer pairs from the text output generated by GPT-4. We first segment the entire output into two parts: the Wikipedia article and Question Answering pairs, identified by the line containing "question", "answer", and "pair". For both chunks, we remove extra symbols like hashes, stars, and consecutive spaces. In the Wikipedia article, we also remove the words "Wikipedia article" at the beginning. For the Question Answering pair chunk, we segment it by line and extract the question and answer by splitting each line using the symbol ":", retaining only the sentences after the ":".

**API**    We accessed GPT-4 via the Azure OpenAI API and collected our entire dataset between the dates of May 24, 2024 and June 5, 2024. We used the gpt-4o-2024-05-13 version of GPT-4 for all of our synthetic data generation.

---

[7]https://github.com/haotian-liu/LLaVA
[8]https://huggingface.co/liuhaotian/llava-v1.5-7b
[9]https://github.com/huggingface/transformers
[10]https://huggingface.co/google/paligemma-3b-mix-224

```
1   Here is a Wikipedia article related to this
2   image:
3
4   [CONTEXT].
5
6   Write question answer pairs which require both the image and the Wikipedia article. The
     question
7   answer pairs should satisfy the following criteria.
8
9   1: The question should refer to the image.
10  2: The question should avoid mentioning the name of the object in the image.
11  3: The question should be answered by reasoning over the Wikipedia article.
12  4: The question should sound natural and concise
13  5: The answer should be extracted from the Wikipedia article.
14  6: The answer should not be any objects in the image.
15  7: The answer should be a single word or phrase and list all correct answers separated
     by commas.
16  8: The answer should not contain 'and', 'or', rather you can split them into multiple
     answers.
```

*Figure 8.* Prompt used to generate only QA pairs for an existing image-context pair using GPT-4.

## I. Details of compute infrastructure used in experiments

We utilized 24 Intel Gaudi2 AI Accelerators to obtain LLaMA-3-70b predictions for our dataset, which were used to create the 'hard' version of our test dataset (Appendix A) and perform LLM evaluation using only questions from our dataset (Appendix E).

For our zero-shot MLLM evaluation and MLLM training experiments, we used an internal linux slurm cluster with Nvidia RTX 3090, Nvidia A6000, and Nvidia A100 GPUs. We used up to 48 GPUs to parallelize various experiments on this cluster. Each parallelized worker was allocated 14 Intel(R) Xeon(R) Platinum 8280 CPUs, 124 GB of RAM, and 1 GPU. The total comptue time for job varied between 6-48 hours depending upon the model, dataset, and evaluation setting.

## J. Topic model details

We removed stop words from context and applied BERTopic (Grootendorst, 2022) to apply categorical TF-IDF on context embeddings created with all-MiniLM-L6-v2 sentence transformer model(Reimers & Gurevych, 2019). We initially reduced the number of topics to 40 using agglomerative clustering. Subsequently, we manually merged semantically related clusters, resulting in the following 25 topics, listed from most to least frequent: general, design, fashion, sports, wildlife, nature, coastal, events, culinary, architecture, transport, pets, structures, space, music, regions, gaming, science, politics, biology, military, postal, entertainment, economics, and religion. The "general" category could not be easily interpreted because it contained a mixture of many different unrelated topics; to improve visual clarity of the figure, it was therefore excluded, and categories representing less than 1% of the dataset were grouped under 'Other' category.

## K. License information

We abide by the licenses and intended uses of all models and datasets which were employed in this study. License information for models and datasets are provided below.

**Models used in our study**    The PaliGemma-3B model is available under the Gemma license. The LLaVA-v1.5-7B is available under the Llama 2 Community License. The LLaVa-v1.6-7B, LLaVa-v1.6-34B, and idefics2-8b are available under the Apache License, Version 2.0. Qwen-VL-7B is available under the Tongyi Qianwen License.

**Existing datasets used in our study**    The WIT dataset is available under the Creative Commons Attribution-ShareAlike 3.0 Unported license. The ViQuAE datset is available under the MIT license. The COCO-Counterfactuals dataset is available under the CC BY 4.0 license. The InfoSeek dataset is available under the Apache 2.0 license.

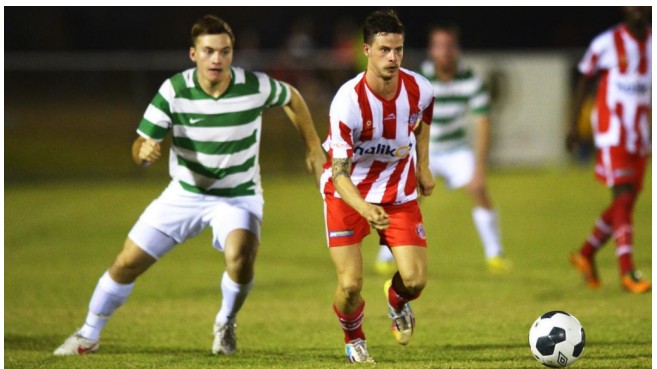

**Context source:** GPT-4
**Context:** In the sport of association football, teams are distinguished by the unique colors and designs of their kits. Kits usually consist of a shirt, shorts, and socks, and teams have both a home and an away version. The home kit is typically in the team's primary colors, while the away kit is in a color that contrasts with their home kit to avoid clashes with other teams. Stripes are a common feature in football kit designs, with many teams opting for vertical or horizontal stripes in two or more colors. Red and white striped kits are popular among several football clubs around the world, a design often associated with clubs like Sunderland AFC and Atlético Madrid. Similarly, green and white horizontally striped kits are also traditional for teams such as Celtic FC. Kits often feature the team logo, the manufacturer's ...

**Context source:** Wikipedia
**Context:** Alice Springs Celtic FC is an Australian soccer club based in Alice Springs, the Northern Territory. The club was one of the first football clubs to be founded in the FFNT Southern Zone Premier League. The club is part of the Football Federation Northern Territory. The club has seen great success in recent years, becoming Premiers in 2015 and Champions in 2016, the club has competed in the annual FFA Cup since 2015 where it reached the Seventh Qualifying Round, eventually losing 5–0 to Darwin Olympic. The club was knocked out in the Alice Springs final in the 2016 edition of the FFA Cup losing 1–0 to Gillen Scorpions and narrowly lost 2-0 in the 2017 edition of the Sports Ministers Cup to Darwin Rovers. At the beginning of the 2017 season prolific striker Joseph Worrall transferred to MPH Vikings ....

*Figure 9.* Comparison of context documents sourced from GPT-4 and Wikipedia for the same Wikipedia image. The context document from Wikipedia contains more entity-specific knowledge, whereas the GPT-4 context document contains more general knowledge about what is depicted in the image.

**Our dataset**   We make our dataset publicly available under the Intel Research & Development License. In addition to the terms of this license, use of our dataset should abide by the OpenAI terms of use.

## L. Additional examples

**Comparison of context documents sourced from GPT-4 and Wikipedia**   A subset of our dataset contains Wikipedia images for which we obtained context documents both from GPT-4 and from Wikipedia (via the image-text associations provided in the WIT dataset). Figures 9, 10, and 11 provide examples of the GPT-4 and Wikipedia-sourced context documents for identical Wikipedia images. The example in Figure 9 shows how context documents obtained from Wikipedia tend to have more entity-specific knowledge, whereas GPT-4 often generates more general knowledge which is related to what is depicted in the image. In Figure 10, the context document generated by GPT-4 is more specific to what is depicted in the image (a vineyard) than what is discussed in the Wikipedia context document (a specific type of wine). Finally, Figure 11 shows how the GPT-4 generated context documents can be longer and more detailed than the Wikipedia context documents which are linked to the image via the WIT dataset.

**Examples of failure cases identified by human annotators**   Figure 12 provides examples of failure cases which were identified by human annotators. As described previously in Section 4.4, the three common categories of synthetic data generation failures identified by human annotators were cases where the question is answerable without looking at the image (Figure 12a), cases where the question is answerable without the context document (Figure 12b), and a single case where insufficient information was provided to answer the question (Figure 12c).

In all three cases depicted in Figure 12, the generated context documents are highly relevant to he image and also contain the answer. However, the question for for example depicted in Figure 12a does not reference the image, and could potentially be answered solely by uni-modal retrieval of the context document based on the question. The question associated with Figure 12b could potentially be answered solely through visual perception capabilities, although the associated context document would increase the chance of generating the correct answer. Finally, the example depicted in Figure 12c is for a

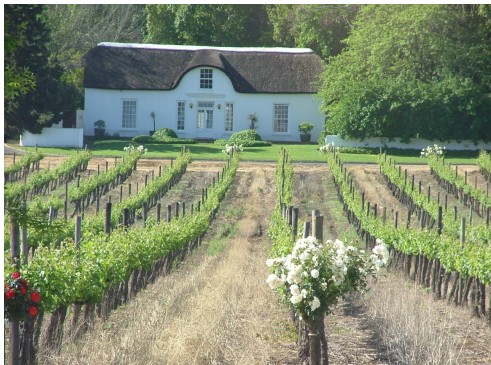

**Context source:** GPT-4
**Context:** A vineyard is an agricultural area where grapevines are grown for the purpose of wine production. The history of viticulture dates back to ancient times, with early evidence of grape cultivation and wine-making discovered in regions such as Mesopotamia, Egypt, and the Mediterranean. Vineyards are typically located on slopes with good sun exposure and well-draining soil, which are ideal conditions for growing grapes. Vineyards are often characterized by rows of trellised vines that maximize sunlight exposure and air circulation, benefiting the health and productivity of the grapevines. The rows of vines are usually planted with companion plants like roses at the end of each row, which serve as an early warning system for diseases that can affect the vines. Wine estates, also known as wineries, are ....

**Context source:** Wikipedia
**Context:** New World wines are those wines produced outside the traditional winegrowing areas of Europe and the Middle East, in particular from Argentina, Australia, Canada, Chile, Mexico, New Zealand, South Africa and the United States. The phrase connotes a distinction between these "New World" wines and those wines produced in "Old World" countries with a long-established history of wine production – most notably, France, Italy, Germany, Spain and Portugal. In 1595 the Spanish Crown banned the establishment of new vineyards in the Americas, but this order was largely ignored. The ban sought to protect Iberian wine from competition by Peruvian wine and can be considered an example of commodity mercantilism. Moreover, the Spanish Crown banned the export of Peruvian wine to Panamá ....

*Figure 10.* Comparison of context documents sourced from GPT-4 and Wikipedia for the same Wikipedia image. The context document sourced from Wikipedia contains knowledge that is less specific to the image than the GPT-4 generated context document.

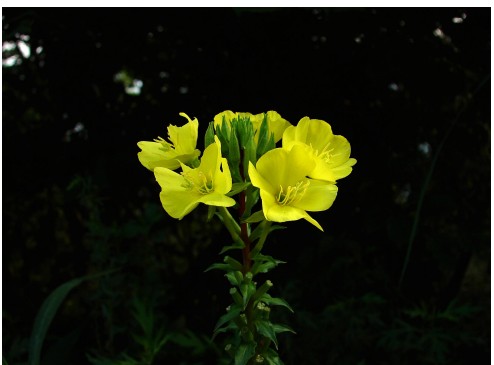

**Context source:** GPT-4
**Context:** The Common Evening Primrose, scientifically known as *Oenothera biennis*, is a biennial plant commonly found in North America. Typically, it grows in less-cultivated areas such as fields and roadsides. The plant is noted for its tall, erect stems that can reach up to 1.5 meters in height. The flowers of the Common Evening Primrose are usually yellow and open in the evening, giving the plant its name. These flowers are known for their sweet fragrance, which attracts pollinators such as bees and moths. The blooming season extends from late spring to early fall, where the flowers are most vibrant. In traditional medicine, the Evening Primrose has been used for various purposes. Native Americans extracted oil from the plant to treat wounds and inflammation. In modern times, ...

**Context source:** Wikipedia
**Context:** Oenothera is a genus of about 145 species of herbaceous flowering plants native to the Americas. It is the type genus of the family Onagraceae. Common names include evening primrose, suncups, and sundrops. They are not closely related to the true primroses.

*Figure 11.* Comparison of context documents sourced from GPT-4 and Wikipedia for the same Wikipedia image. The GPT-4 generated context document is significantly longer and more detailed than the context document sourced from Wikipedia (via the WIT dataset).

question which references a "classical approach", which is not described in the associated context document.

## M. Synthetic Data Quality Control

As discussed in Section 4.5, we thoroughly evaluate our synthetic dataset from various dimensions through manual assessments, complemented by an array of filtering techniques and large-scale automated methods (e.g., grammar checks, bias detection, toxicity screening). We compare our data quality control strategy with other existing approaches that also generate and use synthetic data for training, and present the results in Table 11. As shown in Table 11, we invest significantly more effort in ensuring high-quality data than these prior works.

*Table 11.* Comparisons of Quality Control Methods in Synthetic Datasets (QA: randomly sampled Q&A pairs for manual labeling, grammar: manual grammar review, factual: manual fact-checking using online sources).

| Dataset Name | Human Evaluation Size | Quality Validation / Control Methods |
| --- | --- | --- |
| SK-VQA (ours) | 200 (100 QA, 50 grammar, 50 factual) | heuristic filtering, downstream task evaluation, toxicity/grammar/bias detection with existing model |
| E5 (Wang et al., 2024a) | N/A | downstream retrieval tasks evaluation |
| LLAVA (Liu et al., 2024b) | N/A | downstream task evaluation |
| MiniGPT-4 (Zhu et al., 2024) | N/A | heuristic filtering, downstream task evaluation |
| Owen-VL (Bai et al., 2023) | N/A | downstream task evaluation |
| G-LLAVA (Gao et al., 2023) | N/A | downstream task evaluation |

## N. LLM-as-Judge Evaluation Method Prompts

We present our prompts used in LLM-as-judge methods in Table 12 and Table 13.

*Table 12.* Factuality Evaluation Prompt

**Evaluation Instructions**

You are evaluating the factuality of a description of an image.

1. Look at the image.
2. Carefully read the following description of the image.
3. Score the factuality of the description from 0 to 5:
     - 0 = Completely inaccurate
     - 5 = Fully accurate and matches the image
4. If the score is less than 5, also identify which
specific sentences are inaccurate or misleading and list them in a field called "errors".

**Description:**
[Context comes here]

## O. Limitations and Ethical Considerations

**Limitations**    In this work, we explored the generation of synthetic data from GPT-4 due to its demonstrated state-of-the-art performance in a broad range of multimodal reasoning tasks. While our data generation approach could be used with other MLLMs, we leave the investigation of such applications to future work. Our dataset is limited to English language QA pairs and context documents. While the images in our dataset were collected from a diverse range of sources, they may not be

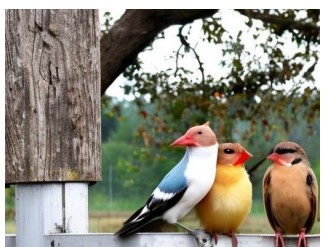

**Image source:** Synthetic (COCO-CFs)
**Context source:** GPT-4
**Context:** Aviary birds are a diverse group of bird species kept in aviaries for observation or conservation purposes. These birds are known for their vibrant plumage and melodic songs, making them popular among bird enthusiasts. Common species found in aviaries include various finches, parrots, and doves, among others….
Popular Species
Some popular species of aviary birds include:
- Gouldian Finches, known for their striking color patterns.
- Zebra Finches, recognized for their distinctive calls.
- Lovebirds, appreciated for their affectionate behavior.
- Cockatiels, valued for their playful nature and ability to mimic sounds.
Importance of Aviaries
Aviaries serve not only as sanctuaries for birds but also as educational resources for the public. They offer a unique opportunity to observe and learn about bird behavior,….
**Q:** Name a popular species of aviary birds characterized by striking color patterns.
**A:** Gouldian Finches

(a) Answerable without image

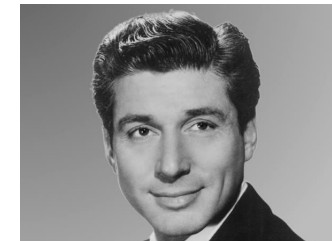

**Image source:** Wikipedia
**Context source:** GPT-4
**Context:** Classic Men's Hairstyles in the 20th Century
During the 20th century, men's hairstyles saw significant transformations that reflected social and cultural changes. One particularly iconic style from the mid-20th century was the neatly combed hairstyle often associated with formality and professionalism.
Mid-20th Century Hairstyle
The hairstyle popular in this era is characterized by its sleek and neatly combed appearance. Men's hair was typically styled with a side part, using gel or pomade to keep the hair in place and achieve a glossy look. This style exuded a sense of refinement and order, aligning with the fashion and cultural values of the time. It was often complemented by wearing formal attire such as a suit and tie.
Fashion and Cultural Influence
The neat and orderly hairstyle represented more than just a grooming choice; it was a reflection of the cultural norms and values of the mid-20th century. This era emphasized …
**Q:** How would you describe the overall appearance of the hairstyle in the image?
**A:** [Sleek, glossy]

(b) Answerable without context

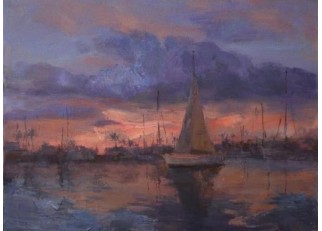

**Image source:** LAION
**Context source:** GPT-4
**Context:** Sunset marine landscapes have been a prominent subject in art for centuries, capturing the serene beauty of the sea as the sun dips below the horizon. These pieces frequently highlight the interplay of light and water, creating reflective surfaces that often enhance the tranquility of the scene. Artists typically use a palette of warm hues such as oranges, pinks, and purples to represent the fading daylight, contrasted against the cooler tones of the approaching night sky and temperate waters. Historical examples of sunset marine landscapes date back to the Romantic era, where the sublime and emotional aspects of nature were emphasized. Renowned painters of the time, including J.M.W. Turner and Claude Monet, used such settings to explore themes of solitude, peace, and the power of nature. Modern interpretations continue to show an appreciation for nautical elements, often depicting sailboats, piers, and harbors against the backdrop of a setting sun. These scenes often evoke a sense of leisure and calm, reminiscent of quiet evenings spent by the water, with minimal human presence to disturb the natural beauty.
**Q:** How does modern interpretation of this theme compare to the classical approach?
**A:** Minimal human presence

(c) Insufficient information to answer

*Figure 12.* Examples of synthetic data generation failures noted by human annotators.

representative of all images domains which might be relevant to users of our dataset.

Due to the scale of our automatically constructed dataset, we are unable to fully validate the accuracy all examples that it contains. We believe that our empirical results provide strong evidence of its quality and usefulness for training MLLMs. While human annotators did not explicitly validate the accuracy of all information contained in evaluated context passages, no obvious cases of hallucination were identified during the annotation process. However, the synthetic nature of the data introduces the possibility that it contains fallacies. Since our primary aim is to train MLLMs to ground generated answers in context documents, we believe such errors pose relatively low risk to the intended use of our dataset. Nevertheless, caution should be exercised when utilizing our dataset, including validation of the performance of any models which are trained on it.

*Table 13.* Quality Evaluation Prompt.

| **Evaluation Instructions** |
|---|
| You are evaluating a visual question answering dataset. |
| **Context (description of the image):** 
 [Context] 

 **Question:** [Question] 

 **Proposed Answer:** [Answer] 

 **Please rate:** 
 1. Relevance of the question to the description (0–5) 
 2. Is the question clearly answerable based on the description? (Yes/No) 
 3. Is the answer factually correct based on the description? (Yes/No) |

**Ethical Considerations**   Our dataset was generated from GPT-4 using the Azure OpenAI API, which includes a content filter for multiple risk categories (e.g., hate speech, fairness, sexual language, violence). As this filter automatically removes potentially offensive content that is generated by GPT-4, we believe that the likelihood of our dataset containing such content is relatively low. However, content filtering models are not infallible and we are unable to manually inspect our entire dataset for the presence of offensive content due to its large scale. It is also possible that potentially harmful biases possessed by GPT-4 which do not trigger content filters are reflected in our dataset. Users should carefully consider these risks relative to the benefits of our synthetic dataset before deploying systems which are built using it.

