# OpenReview forum: "SK-VQA: Synthetic Knowledge Generation at Scale for Training Context-Augmented Multimodal LLMs"
_ICML.cc/2025/Conference — ICML 2025 oral_

### Official Review · Reviewer_eTjf · 2025-03-06

**Overall Recommendation:** 4

**Summary:**

The paper proposes a dataset for visual question answering with external knowledge or for evaluating MLLM + RAG systems.

The dataset is constructed by using images from multiple datasets as seed images, and then writing context for those images using GPT-4o or using paired Wikipedia context when available, and then writing questions based on those contexts. Human analysis is done on the generated question-answer pairs. A number of quality control steps are performed.

The experiments section describes a series of experiments where transfer is measured from the proposed dataset to other datasets.

The proposed dataset is substantially larger than existing datasets, and training on the proposed dataset transfers well to other datasets. The dataset is of comparable difficulty to existing datasets w.r.t to both human and model evaluation. Quality control checks suggest that the level of noise in the dataset are low.

**Claims And Evidence:**

Yes.

**Essential References Not Discussed:**

None that I am aware of.

**Experimental Designs Or Analyses:**

I checked all of $\S5$. I do not see any issues.

**Methods And Evaluation Criteria:**

Yes.

**Other Comments Or Suggestions:**

The authors should have a "-lite" split of their dataset. This dataset is large, and to make it accessible to the community, you should prepare a smaller evaluation split of the dataset that could be evaluated on in a smaller amount of time.

**Other Strengths And Weaknesses:**

The main strength is that this is a high quality dataset with a well-thought out design that appears to be more effective as a source of training data than previous datasets. The main weakness is that conceptually, it seems to offer nothing new (other than being larger + higher quality) than existing datasets.

Additional strengths:
- The dataset contains nearly 50% more questions than the next largest dataset.
- In addition to being larger, it is substantially more diverse (11x unique questions vs EncylopedicVQA).
- There is a human evaluation performed and all humans perform similarly on the dataset.

Additional weaknesses:
- Open-ended VQA datasets are known to contain high levels of noise. Specifically, for some questions the annotated answer might not be the only correct answer, or some questions might be ill-posed. I did not see any analysis done on how many model errors are the result of possibly noisy questions vs a genuine error. This could be done by evaluating a model on a subset and looking at a few errors (let's say 50-100).
- I don't see any evaluations of frontier models like GPT-4o on this dataset. If this dataset is generated by GPT-4o, does that mean it is already "solved" by GPT-4o and cannot be used to evaluate GPT-4o. This is likely not the case for something like EncVqa, for which AFAICT even frontier models (at the time of evaluation) perform poorly.

**Questions For Authors:**

1. What was your motivation for this work? In particular, can you point out the problems with InfoSeek or EncylopedicVQA that you were trying to solve with this paper? Are there new applications that SK-VQA enables that were nontrivial to do with EncyclopedicVQA or InfoSeek? Concrete examples would be helpful. If answered convincingly, I will raise my rating.
2. I don't see any evaluations of frontier models like GPT-4o on this dataset. If this dataset is generated by GPT-4o, does that mean it is already "solved" by GPT-4o and cannot be used to evaluate GPT-4o? This is likely not the case for something like EncVqa, for which AFAICT even frontier models (at the time of writing) perform poorly, though they only evaluate on GPT-3. Should frontier models only be evaluated on the WiT split of the dataset?

**Relation To Broader Scientific Literature:**

External knowledge visual question answering datasets started off small, with datasets like OK-VQA and A-OKVQA. These datasets have been superseded by more recent, larger datasets that are harder, like InfoSeek and EncyclopedicVQA. The proposed dataset is larger and appears to be higher quality than InfoSeek and EncylopedicVQA. I'm not sure if it is conceptually any different than EncyclopedicVQA and InfoSeek.

**Theoretical Claims:**

There are no proofs or theoretical claims.

---

> ### Author Rebuttal · Authors · 2025-04-01
>
> We sincerely appreciate your thoughtful review and your recognition of the dataset’s quality, thoughtful design, large scale, and strong transfer performance. We have carefully addressed your comments and concerns as follows:
>
> > **...did not see any analysis done on how many model errors are the result of possibly noisy questions vs a genuine error...**
>
> > **...I don't see any evaluations of frontier models like GPT-4o on this dataset...***
>
> During the rebuttal period, we evaluated GPT-4o on the SK-VQA test set, and it achieved a score of 58.9%. We also conducted a manual error analysis on randomly sampled 50 examples that were marked incorrect by automatic evaluation. Our findings show that:
>
>
> - 40% were genuine model errors
> - 30% were partially correct
> - The remaining 30% were actually correct.
>
> We will add these additional GPT4o’s results in our final version.
>
>
> > **...you should prepare a smaller evaluation split of the dataset that could be evaluated on in a smaller amount of time.**
>
> We fully agree with the reviewer’s suggestion. To improve accessibility and encourage broader adoption, we will release a “-lite” evaluation split alongside the full dataset. This smaller subset will be designed to run efficiently on limited compute while preserving task diversity.
>
> > **What was your motivation for this work? In particular, can you point out the problems with InfoSeek or EncylopedicVQA that you were trying to solve with this paper? Are there new applications that SKVQA enables that were nontrivial to do with EncyclopedicVQA or InfoSeek? Concrete examples would be helpful...**
>
> We appreciate this opportunity to clarify our motivation and contributions.
>
> Our goal was to address key limitations of existing KB-VQA datasets such as InfoSeek and Encyclopedic-VQA, which include:
>
> - Narrow image coverage: These datasets mostly include images that can be linked to Wikipedia entities (e.g., landmarks, animals), excluding everyday or abstract visuals. In contrast, SK-VQA includes images from LAION, Wikipedia, and COCO-Counterfactuals, enabling coverage of open-domain, synthetic, and artistic images.
> - Low question diversity: InfoSeek uses templated QA generation (e.g., “What is the capital of X?”), leading to <1% unique questions. SK-VQA uses GPT-4 to generate both context and QA pairs together, resulting in ~96% unique questions with richer phrasing (Table 2).
> - Limited knowledge types: Prior datasets focus on entity-centric facts. SK-VQA includes a broader range of topics (25 identified via topic modeling; Fig. 4), including art, cultural events, sports, and fashion.
>
> Some concrete examples:
>
> - Figure 1 (main paper): Shows a question about the Golden Globe Awards hosted at the Beverly Hilton — a cultural-event-centric question that would be unlikely in InfoSeek or Encyclopedic-VQA due to lack of coverage.
> - Figure 2: Demonstrates questions such as “What characteristic helps this breed adapt to cold water?” (about Labrador Retrievers) — combining visual traits with world knowledge, beyond simple object labels.
> - Appendix Figure 10: Compares a GPT-4 generated context about vineyards to a Wikipedia context about New World wines. The synthetic version better aligns with the image and supports diverse, image-grounded questions.
>
> New Applications Enabled by SK-VQA:
>
> - Multimodal RAG Training & Evaluation: SK-VQA includes paired image, context, and QA for over 2M examples — a scale not offered in existing datasets — enabling training of models that retrieve and reason over context.
> - Fully-synthetic training and counterfactual reasoning: The inclusion of COCO-CFs and GPT-4 generated knowledge allows SK-VQA to support training and testing on hypothetical, non-real-world scenarios.

---

### Official Review · Reviewer_GcyS · 2025-03-11

**Overall Recommendation:** 4

**Summary:**

This paper provides and analyzes a new dataset called SK-VQA, which is a large-scale dataset designed to train multimodal language models for knowledge-based visual question answering with context augmentation.

The authors’ motivation is that existing datasets for this specific task do not cover large and diverse enough topics and questions. They leverage GPT-4 to produce synthetic data, resulting in a dataset with greater question diversity and broader knowledge coverage compared to previous resources. Evaluation demonstrates the proposed dataset could serve as a challenging benchmark and an effective training tool.

**Claims And Evidence:**

There are three major claims made in this paper.
1. Introduction of SK-VQA as a large-scale and diverse synthetic multimodal dataset for context-augmented KB-VQA: This claim is supported by its size when compared to other datasets.
2. SK-VQA exhibits greater question diversity compared to previous datasets: It is supported by Table 2. It has significantly more questions than other KB-VQA datasets like InfoSeek and Encyclopedic-VQA.
3. SK-VQA presents challenges on existing models, especially for zero-shot: This result should be supported by Figure 5. However, it is not clear why the results of training and testing with SK-VQA are missing in the figure. The author could split the proposed dataset into two splits and provide the results.

**Essential References Not Discussed:**

The paper might lack some recent models, like Qwen VL2, Ovis, InternVL, Molmo, etc.

**Experimental Designs Or Analyses:**

1. The finetuned results of different datasets are not clear. In table 5, I feel the author is using split training and test sets of InfoSeek during training and test stages while using the same proposed SK-VQA both for training and testing. In that case, the author should make it clear in the paper.

**Methods And Evaluation Criteria:**

1. The proposed data synthesis method is targeted at addressing the data scarcity problem. Although the motivation is reasonable, the generation of context documents is questionable. The quality of the generated context documents is not well assessed in terms of correctness, diversity, and completeness.

2. Another question is since both the context documents and QAs are all generated with GPT-4, I am worried that the quality of the dataset might not keep up with the current advanced multimodal LLMs. As little human effort is included in the loop during data synthesis, I wonder if the GPT-4 generated dataset is qualified to evaluate other models that already achieve better performance than GPT-4 on most benchmarks.

3. The evaluation results look comprehensive while lacking some recent models, like Qwen-VL2, Ovis, and Molmo, etc. It would be more informative to readers if these methods could be included.

**Other Comments Or Suggestions:**

What is the release plan of the proposed dataset?

**Other Strengths And Weaknesses:**

Weakness:
1. One major concern is about how the accompanied document being generated in the proposed dataset. As stated in Section 3.1, the context document is generated with QA pairs from GPT-4 at the same time. However, as the context documents are generated from a single model using the same prompt, they might lack diversity in both the knowledge and the writing styles. This might make the synthesized dataset 'not real' compared to other multimodal RAG datasets.

Similarly, I am not conviced by the argument between L194 - 196. From my point view, the consistancy can also be guaranteed if we first prepare the context document from web or other resouces and then ask GPT-4 to prepare QA pairs based on the document.

2. The quality assessment of the generated document is not comprehensive. The author discuss 'no obvious cases of hallucination were identified' in L856 of their supplemenraty. However, the correctness of the generated document is not shown or hard to evaluate based on the propsoed dataset creation pipeline. Besides, it is also important to discuss the knowledge coverage of the generated document.

**Questions For Authors:**

I have no other questions.

**Relation To Broader Scientific Literature:**

The dataset would be a choice of evaluating context-aware VQA models and might benefit the multimodal LLM society.

**Theoretical Claims:**

The paper is about proposing a dataset. There is no theoretical claim.

---

> ### Author Rebuttal · Authors · 2025-04-01
>
> We sincerely appreciate your detailed review and your recognition of the dataset’s scale, diversity, and its potential to benefit context-aware multimodal research. We have addressed your concerns as follows:
>
>
> > **it is not clear why the results of training and testing with SK-VQA are missing in the figure 5**
>
> Figure 5 focuses only on out-of-domain generalization, where models are trained on one dataset and evaluated on the test sets of other datasets. This is why we did not include results where both training and testing are done on SK-VQA (i.e., in-domain performance).
>
> The in-domain performance—where training and testing are on SK-VQA—is provided in Table 6 (Appendix A) for completeness. We intentionally separated in-domain and out-of-domain results to better highlight the generalization ability of each dataset.
>
> > **The quality of the generated context documents is not well assessed in terms of correctness, diversity, and completeness.**
>
> > **..the correctness of the generated document is not shown...**
>
> During the rebuttal time, we have added a new experiment based on LLM-as-judge (GPT-4o) to analyze the quality of the dataset. Specifically, we asked the model to check the factuality, question relevancy, question answerable, and answer correctness.
>
> - Factuality (0 = Completely inaccurate to 5 = Fully accurate and matches the image): the average score is 4.6, with 87.5% cases scored of “5”.
> - Question relevancy (0 = not relevant to image and context to 5 = relevant to image and context): the average score is 4.9, with 92.0% is a score of 5.
> - Question Answerable(Yes/No): 99.6% questions are answerable.
> - Answer correctness (Yes/No): 90.7% questions are answerable.
>
> Additionally, we performed an additional 100 human analysis as in section 4.4, the results are consistent with the analysis in the Table 3 of the paper, reinforcing the reliability and representativeness of our evaluation. These new experiments plus the analysis in our paper for correctness (Table 3 for Human analysis and Section 4.4.2 for grammar), diversity (Table 2), completeness (above new LLM-as-judge analysis), and Bias and Toxicity (Section 4.2.2), show that SK-VQA undergoes richer validation, both automated and human, than most existing benchmarks (see Appendix L for a detailed comparison).
>
> > **I wonder if the GPT-4 generated dataset is qualified to evaluate other models...lacking some recent models, like Qwen-VL2, Ovis, and Molmo...**
>
> During the rebuttal, we evaluated more recent and powerful VLMs, including Qwen-2.5-VL (3B/7B/32B/72B) and Ovis (1B/2B/4B/8B/16B/34B). On SK-VQA, these models achieved:
>
> - Qwen-2.5-VL: 53.74, 49.26, 52.08, 49.09
> - Ovis: 32.25, 44.54, 50.55, 50.36, 52.36, 55.2
>
> For comparison, on ViQuAE, the same Ovis models achieved:
>
> - Ovis: 39.50, 67.09, 49.38, 57.96, 72.77, 67.03
>
> These results show that SK-VQA is a more challenging dataset compared to the existing ones.
>
> > **lack diversity in both the knowledge and the writing styles ... consistancy can also be guaranteed if we first prepare the context document from web or other resouces ...**
>
> We address the two points as follows:
> On diversity of generated contexts: We designed our pipeline and prompts as open-ended to promote both linguistic and knowledge diversity. As shown in Table 2, SK-VQA exhibits significantly higher diversity in POS patterns, vocabulary, and question structure than prior datasets. Figure 4 further shows that our context documents span 25+ knowledge domains, based on unsupervised topic modeling. Additionally, the use of diverse image sources (e.g., LAION, Wikipedia, COCO-CFs) ensures a wide range of visual prompts for generation, resulting in stylistic and content variation. The strong zero-shot difficulty across models (Table 4) also supports the idea that the data is not overly templated or repetitive.
>
> On generating QA and context together: We agree that it is possible to first retrieve real documents and then generate QA pairs. However, our method generates context and QA jointly in one step, which allows us to explicitly control key constraints — such as ensuring the answer is only in the context (not the image), that object names are avoided, and that reasoning is required. This level of alignment is difficult to achieve when using unstructured web data, where the context may not be tailored to support the desired QA types.
>
> > **...it is also important to discuss the knowledge coverage...**
>
>  We applied topic modeling and identified 25 distinct domains (Figure 4), showing broad topical diversity beyond entity-specific facts.
>
>  > **What is the release plan of the proposed dataset?**
>
>  We will publicly release the full SK-VQA dataset, along with the code, upon acceptance.

---

### Official Review · Reviewer_nbTf · 2025-03-14

**Overall Recommendation:** 4

**Summary:**

This paper presents a large-scale synthetic dataset containing over 2 million visual questions with answers that require information from associated context. The images used in this dataset are from a hybrid of synthetic images from COCO-CFs and real images from Wikipedia and LAION, while the context and answers are generated from GPT-4. In the evaluation, they illustrate that this dataset can be used as a challenging benchmark of KB-VQA models and can also be effectively used as training data for multimodal context-augmented generation.

**Claims And Evidence:**

1. One concern I have is regarding the quality of the question-answer pairs. It seems that the only quality control in this paper regarding the validity of question-answer pairs is the human evaluation. While it is conducted on a very small scale (100 QA pairs). Therefore, it may be questionable whether the dataset can be used as a reliable benchmark to evaluate the KB-VQA models.

**Essential References Not Discussed:**

N/A

**Experimental Designs Or Analyses:**

Most of the experimental designs and analyses are sound to me. One issue I found is related to the impact of using generation source and real source for Wiki-based images (Table 5). It is not clear to me whether it is a fair comparison because from Table 1, we can find that the number of synthetic contexts is much larger than the number of real contexts from Wikipedia. So it is not clear to me whether the advantage is coming from the quality of the synthetic context and QA pairs, or just that the number of the contexts is increased.

**Methods And Evaluation Criteria:**

The proposed method is, in general, technically sound. By synthetically generating context and QA pairs, it can potentially increase the diversity of data when training the KB-VQA models. The evaluation of using different data sources at a similar scale as the training set and test on other datasets is also legitimate.

Another evaluation that is currently missing is the ablation of using different image sources. It is not clear what roles different image sources play in either training or evaluation. Especially in this paper, there are also some synthetic images. It would be meaningful to see some analysis in this direction.

**Other Comments Or Suggestions:**

The title of Section 4.4 is *Human Evaluation*, while some part of Section 4.4.2 *Additional Dataset Quality Evaluations* are automatic evaluation using LanguageTool. The authors may consider using other names to avoid confusion.

**Other Strengths And Weaknesses:**

This paper presents an interesting way that additionally focuses on generating the context and then the question answer pairs, as well as using synthetically generated images.

**Questions For Authors:**

The context in WIT dataset is organized at different levels, for example, caption, paragraph or the whole Wikipedia page. What kind of contexts are you using to compare with the synthetic contexts?

**Relation To Broader Scientific Literature:**

This proposed synthetic data generation method can be potentially adopted by other KB-VQA work to enhance the training data or as a challenging benchmark for evaluation.

**Theoretical Claims:**

N/A

---

> ### Author Rebuttal · Authors · 2025-04-01
>
> Thank you for your thoughtful and constructive review. We're grateful for your recognition of the strengths of our approach — particularly the scale and diversity of the dataset, our use of varied image and context sources, and the overall soundness of our methodology. We have addressed your concerns in detail as follows:
>
> > **One concern I have is regarding the quality of the question-answer pairs...**
>
> We appreciate the reviewer’s concern regarding the sample size of our human evaluation. We would like to highlight that our initial evaluation of 100 QA pairs already exceeds the human evaluation effort in prior synthetic dataset works (see Appendix L for a detailed comparison).
>
> To further address this concern, during the rebuttal period we conducted an additional human evaluation on 100 new QA pairs using the same methodology described in Section 4.4. The results remained consistent with the original analysis, with a mean accuracy of 77.0%, reinforcing the reliability and representativeness of our evaluation.
>
> - For factuality, we asked the model to score the factuality of the description from 0 to 5: 0 = Completely inaccurate, 5 = Fully accurate and matches the image. The result shows the average score is 4.6, with 87.5% cases scored of “5”.
> - For the question relevancy, we asked the model to score the relevance of the question to the description and image (0–5). The result shows the average score is 4.9, with 92.0% is a score of 5.
> - For the question answerable, we ask the model is the question clearly answerable based on the description? (Yes/No). The result shows 99.6% questions are answerable.
> - For the answer correctness, we ask “Is the answer factually correct based on the description? (Yes/No)”. The result shows 90.7% questions are answerable.
>
> These results, combined with our original and extended human evaluations, automated grammar checks (LanguageTool), fact-checking (via manual validation), bias/toxicity screening, and strong downstream task performance across multiple benchmarks, collectively demonstrate the high quality and utility of our dataset.
>
> > **Another evaluation that is currently missing is the ablation of using different image sources...**
>
> We thank the reviewer for the suggestion. In fact, we have already included this analysis in Table 5 and Table 8 of our paper.
>
> In particular, Table 5 shows that models trained on synthetic images (COCO-CFs) paired with GPT-4 context perform on par with or better than those trained on real images (e.g., from Wikipedia). Table 8 further explores how filtering methods interact with image sources, showing that certain sources (e.g., LAION vs. Wiki) generalize differently across downstream tasks.
>
> These results suggest that diverse image sources — including synthetic ones — contribute positively to generalization, and that combining them can be more effective than relying on a single source. We will make this clearer in the final version.
>
> > **...it is not clear to me whether the advantage is coming from the quality of the synthetic context and QA pairs, or just that the number of the contexts is increased.**
>
> To ensure a fair comparison, all results in Table 5 were obtained using equal-sized subsets (downsampled to 200K training samples per setting). We will make this clarification more explicit in the final version of the paper.
>
> > **The title of Section 4.4 is Human Evaluation...consider using other names to avoid confusion.**
>
> Thank you for the helpful suggestion. In the final version, we will keep 4.4. As Human Evaluation, and split the 4.4.2 to a new section titled “Automatic Evaluation”.
>
> > **The context in WIT dataset...What kind of contexts are you using to compare with the synthetic contexts?**
>
> Thank you for pointing this out. For Wikipedia-based contexts, we use the paragraph-level context associated with each image from the WIT dataset. We will make this more clear in our final version of the paper.

---

> > ### Comment · Reviewer_nbTf · 2025-04-03
> >
> > Thanks for the authors' response. I would like to further clarify my questions regarding the role of different data sources. Because the proposed data consists of multiple resources, besides training the model with each of the individual resources to demonstrate the differences between different resources, I would also like to see what is the impact of each resource on the overall data. One example would be similar to the ablation study, in which we remove each resource from the overall data and train the model from the remaining data to evaluate the impact of this resource on the overall data.
> >
> > Similarly, regarding using the synthetic data for evaluation, it is also nice to have some analysis that can break into each resource. For example, are there certain resources in general easier or difficult than others? These analyses will help us better understand the role of each image and context source.

---

> > > ### Author Response · Authors · 2025-04-07
> > >
> > > Dear reviewer, we sincerely appreciate your insightful suggestion. In response, we have conducted an evaluation of the performance of five models on different subsets of our dataset, divided based on the source and type of image content. The results demonstrate that each subset presents a distinct level of difficulty. Specifically, we observe a consistent increase in difficulty across the following order: WiT (Wiki content), WiT- (GPT-4 generated content), LAION, and Coco-CF.
> > > We hypothesize that the WiT (Wiki) subset is the easiest because large language models are likely to have been trained on a substantial amount of Wikipedia content, making this subset more familiar and easier to answer. In contrast, the Coco-CF subset includes counterfactual image and GPT-4 generated content pairs that are largely out-of-distribution relative to the training data of these models, thus presenting the highest degree of difficulty.
> > > These findings highlight the diversity of our dataset and underscore the importance of incorporating varied content sources—especially those beyond Wikipedia-based images, which are predominantly used in many existing knowledge-based VQA datasets—in the construction of SK-VQA. We will include this analysis in the final version of the paper.
> > >
> > > | Model             | LAION | WiT(GPT-4) | WiT(Wiki) | Coco-CF |
> > > |------------------|-------|------------|-----------|---------|
> > > | LLaVA-v1.5-7B     | 40.99 | 44.35      | 50.45     | 41.4    |
> > > | LLaVA-v1.6-7B     | 46.68 | 48.9       | 54.8      | 46.85   |
> > > | Qwen-VL-7B        | 42.55 | 42.45      | 47.6      | 41.6    |
> > > | LLaVA-v1.5-13B    | 40.42 | 41.5       | 50.85     | 39.4    |
> > > | LLaVA-v1.6-13B    | 45.57 | 46.5       | 56.25     | 43.5    |

---

### Official Review · Reviewer_jWqo · 2025-03-14

**Overall Recommendation:** 4

**Summary:**

This paper introduces SK-VQA, a dataset with over 2 million question-answer pairs associated with context documents for training multimodal language models in knowledge-based visual question answering. Using GPT-4, the authors generated context documents and diverse QA pairs for images from varied sources, creating a dataset with 11× more unique questions than existing resources. Their experiments show that SK-VQA serves as both a challenging benchmark and effective training resource, with models trained on it demonstrating superior generalization in context-augmented settings compared to models trained on other datasets. This addresses a critical limitation of current multimodal LLMs which aren't designed for context-augmented generation in knowledge-intensive tasks.

**Claims And Evidence:**

The paper's central claims about SK-VQA's size, diversity, and performance improvements are well-supported by quantitative evidence. The dataset metrics showing 11× more unique questions than comparable datasets are documented in Table 2, while the performance advantages of models trained on SK-VQA are consistently demonstrated across multiple experiments in Figure 5 and Table 4.

However, there is one major limitation: the human evaluation covered only 100 QA pairs (0.005% of the dataset), raising questions about the representativeness of the evaluated samples and therefore overall quality of the dataset.

**Essential References Not Discussed:**

NA.

**Experimental Designs Or Analyses:**

The zero-shot and fine-tuning evaluations use appropriate metrics and multiple model sizes, strengthening validity. The comparative analysis against InfoSeek, Enc-VQA, and ViQuAE provides the necessary benchmarking context.

However, the RAG experiments use only one model architecture (PaliGemma-3B), limiting generalizability claims across architectures. Also, when testing with LLaMA-3-70b to create a "hard" subset, the authors don't clearly establish what percentage of questions are answerable by looking only at context (only provided the final number of samples), making it difficult to assess the true multimodal reasoning requirements of the dataset. Is it the case that only $2,853$ out of 2 million can be answered just using the context?

**Methods And Evaluation Criteria:**

Yes. Using GPT-4 to generate synthetic QA pairs and context documents addresses the scarcity of suitable training data, while the filtering techniques (IR and CAP) help ensure data quality. The evaluation metrics, such as BEM and exact match, are standard. The evaluation framework is comprehensive, examining both zero-shot performance on existing benchmarks and fine-tuning outcomes across multiple models, including out-of-domain generalization which is particularly relevant for real-world applications. The RAG experiments simulate practical use cases where retrieved knowledge must be integrated with visual information. The comparison against existing datasets (InfoSeek, Enc-VQA, ViQuAE) provides meaningful context.

**Other Comments Or Suggestions:**

Table 6 caption misspelled: "semantic matric".

**Other Strengths And Weaknesses:**

Strength: the diversity of the dataset is much greater than prior work -- over 96% of the questions in SK-VQA are unique, an 11x improvement than Enc-VQA; the questions in this work also have a greater number of unique POS sequences, total vocabulary size, and mean word length.

Weakness: No assets were provided in the submission. The reviewer recommends the authors release the datasets and codebase upon acceptance.

**Questions For Authors:**

From Lines 327 (left) - 287 (right), "Factual accuracy is not a primary concern...  as its main purpose is to train MLLMs to effectively
utilize long contexts for VQA ... specifically, we ask a native speaker to fact-check 50 QA pairs and supporting evidence in context documents using online sources. 86% were verified as factual, 4% were non-factual, 2% were partially factual, and 8% could not be determined due to a lack of available information."

Could the authors further explain why factual accuracy is not a primary concern as a QA dataset? Noisy labels may undermine the performance from training / fine-tuning the MLLMs, and the 86% factual accuracy may hinder the practical usage of this dataset.

**Relation To Broader Scientific Literature:**

The prior KB-VQA datasets seem to already be included in the paper.

**Theoretical Claims:**

No theoretical claims are provided in the paper.

---

> ### Author Rebuttal · Authors · 2025-04-01
>
> We sincerely appreciate your recognition of the strengths of our work, including the dataset’s scale and diversity, the quality-controlled generation process, and the robustness of our experimental design. We have carefully addressed all your concerns as follows:
>
> > **However, there is one major limitation: the human evaluation covered only 100 QA ...**
>
> We appreciate the reviewer’s concern regarding the sample size of our human evaluation. We would like to highlight that our initial evaluation of 100 QA pairs already exceeds the human evaluation effort in prior synthetic dataset works (see Appendix L for a detailed comparison).
>
> To further address this concern, during the rebuttal period we conducted an additional human evaluation on 100 new QA pairs using the same methodology described in Section 4.4. The results remained consistent with the original analysis, with a mean accuracy of 77.0%, reinforcing the reliability and representativeness of our evaluation.
>
> - For factuality, we asked the model to score the factuality of the description from 0 to 5: 0 = Completely inaccurate, 5 = Fully accurate and matches the image. The result shows the average score is 4.6, with 87.5% cases scored of “5”.
> - For the question relevancy, we asked the model to score the relevance of the question to the description and image (0–5). The result shows the average score is 4.9, with 92.0% is a score of 5.
> - For the question answerable, we ask the model is the question clearly answerable based on the description? (Yes/No). The result shows 99.6% questions are answerable.
> - For the answer correctness, we ask “Is the answer factually correct based on the description? (Yes/No)”. The result shows 90.7% questions are answerable.
>
> These results, combined with our original and extended human evaluations, automated grammar checks (LanguageTool), fact-checking (via manual validation), bias/toxicity screening, and strong downstream task performance across multiple benchmarks, collectively demonstrate the high quality and utility of our dataset.
>
> > **Also, when testing with LLaMA-3-70b ... what percentage of questions are answerable ...**
>
> Thank you for raising this point. To clarify: we applied LLaMA-3-70B-Instruct to QA pairs from the SK-VQA testing set, providing only the context document and question, without access to the image. Among these 26.5% are answerable, and we use the rest 73.5% which are answered incorrectly by the model as the hard subset. We will add this percentage in the paper.
>
> > **... The reviewer recommends the authors release the datasets and codebase upon acceptance.**
>
> We fully agree, and we confirm that we will release the full SK-VQA dataset, as well as the code, upon acceptance of the paper.
>
> > **Table 6 caption misspelled: "semantic matric".**
>
> Thank you for pointing this out. We will correct the typo in Table 6 and change "semantic matric" to "semantic metric" in the final version.
>
> > **... why factual accuracy is not a primary concern ...**
>
> We would like to clarify that the term “factual accuracy” in our paper refers to the alignment of the context document with real-world facts—not the correctness of the answer with respect to the context. And this alignment with real-world knowledge is not necessary for our task because we aim to teach models to effectively utilize long multimodal contexts for grounded reasoning, rather than memorizing the context.
>
> That said, we still conducted fact-checking via manual verification as described in the paper. In addition, during the rebuttal, we applied the LLM-as-judge approach to assess factuality, question relevance, and answer correctness (the results are mentioned in the previous answer), all of which further support the high quality and utility of our dataset.

---

> > ### Comment · Reviewer_jWqo · 2025-04-04
> >
> > The reviewer sincerely thanks the authors for their great efforts. The responses address my concerns, and I am increasing the rating to 4.

---

> > > ### Author Response · Authors · 2025-04-07
> > >
> > > we appreciate reviewer's recognition and sincerely thanks your valuable and insightful comments which improve our work.

---

### Decision · Program_Chairs · 2025-05-01

**Decision:**

Accept (oral)

**Comment:**

After the rebuttal, all reviewers have raised their ratings to Accept (4), and all concerns are addressed by the authors. It is a clear accept paper, and it would be better to incorporate all rebuttal discussions into the camera-ready version.